# Impact of Geopolitical Risk on the Information Technology, Communication Services and Consumer Staples Sectors of the S&P 500 Index

**Gerard Atabong Fossung, Vasileios Chatzis Vovas and A. M. M. Shahiduzzaman Quoreshi \***

Department of Industrial Economics, Blekinge Institute of Technology, 371 41 Karlskrona, Sweden;
gerard.fossung@gmail.com (G.A.F.); vasileios.chatzis.vovas@outlook.com (V.C.V.)

\* Correspondence: shahiduzzaman.quoreshi@bth.se; Tel.: +46-455−385-638

**Abstract:** We investigate the effect of geopolitical risk on the returns of firms in the Information Technology, Communication Services, and Consumer Staples sectors within the S&P 500 index. We use the event study methodology and perform more than 17,000 regressions to provide empirical evidence at sector level that geopolitical risk leads to different responses across these three sectors. The response of the Information Technology sector is negative for all event windows under study, except the one spanning 10 days prior to the geopolitical event and 10 days after. The Communication Services sector has positive returns as a result of geopolitical events for all event windows, except the one from the geopolitical event date and 5 days after. The Consumer Staples sector shows a negative impact on geopolitical risk for all event windows except the one from the geopolitical event date and 5 days after, demonstrating a negative correlation to the Communication Services sector.

**Keywords:** geopolitical risk; event study; S&P 500 index; information technology; communication services; consumer staples



## 1. Introduction

"Geopolitics covers a diverse set of events with a wide range of causes and consequences, from terrorist attacks to climate change, from Brexit to the Global Financial Crisis" (Caldara and Iacoviello 2018). Adverse geopolitical events and threats often result in uncertainties or risks on global economies, local economies, as well as general financial markets such as the stock exchange. Campbell et al. (2012) asserted that uncertainty plays a vital role in financial economics it has an impact on the behavior of investors and market prices. Carney (2016) identified geopolitical uncertainty, economic uncertainty, and policy uncertainty, as the "uncertainty trinity" that influences economic performance. Geopolitical risk (GPR) exists not only because of the risks associated with the realization of (adverse) events but also due to the escalations thereof. Caldara and Iacoviello (2018) found that while the realization of adverse geopolitical events leads to smaller economic effects because the uncertainty is usually addressed and resolved, the shocks of geopolitical threats are usually protracted, leading to a rise in uncertainty and adverse economic activity. Their study also revealed that in the U.S., as well as around the world, stock returns respond asymmetrically to the threats and realizations of geopolitical events. Derousseau (2018) established that the defense industry, the oil industry, and the consumer industry are three industries where stocks should be owned during risky times. On the global economic front, both the World Bank (2020) and Dimitrijevic et al. (2019) concurred that political and trade tensions erode confidence and investment plans, as well as stifle expectations of global economic growth. Lastly, according to Dimitrijevic et al. (2019), the U.S.–China trade and technology disputes, the proxy wars in the Middle East, U.S. President Trump's impeachment hearings in the U.S., and the Brexit imbroglio are drivers of global tensions that lead to slower global economic growth, with the U.S.–China relationship as the main catalyst of this economic slowdown.

Wade and Lauro (2019) argued that geopolitical risk is on the rise, and in the future it will be fueled by two factors which are: the rise of China, and the rise of populism. The authors also identified the uncertainty surrounding the U.S.–China trade tension and the decision of the U.K. to leave the European Union as recent events to which close attention needs to be paid regarding geopolitical risk. Furthermore, Wade and Lauro (2019) analyzed the performance of "safe" and "risky" assets during periods of high GPR and observed that the portfolio of "safe" assets delivered higher risk-adjusted returns than those of the "risky" assets in four out of five periods considered. As a result, they advised investors to consider GPR, as it benefits portfolio performance through diversification of assets from "risky" to safe-haven assets when GPR becomes high.

A lot of studies have been performed by researchers on the impact that geopolitical risk has on stock volatility within some economic industries. For example, Chkili et al. (2014) and Antonakakis et al. (2017) both studied the effect of GPR on the oil industry, while Caldara and Iacoviello (2018) analyzed and summarized the GPR effects on general market price indexes at the industry-level (49 industries) and across 17 countries between 2005 and 2018. Caldara and Iacoviello (2018) used forecasting regressions and vector autoregressive (VAR) models, and showed that a shock to geopolitical risk impacts stock markets. In contrast, our study investigates the GPR impact on the stock returns of firms on the S&P 500 index (in the U.S.) at sector level (132 sector-level firms from three different sectors) between 1962 and 2020, using a time-series event study methodology (MacKinlay 1997; Campbell et al. 2012).

To provide an easy understanding of the structure of the industries and sectors in the U.S. S&P 500 index to the reader, we capture the breadth, depth, and evolution of its industry sectors in Figure 1 below. Alternatively, every S&P 500 index sector and their underlying industries can clearly be seen in Ross (2020). In particular, the S&P 500 index consists of 505 firms, spanning 11 sectors, 126 industries (Slickcharts 2020).

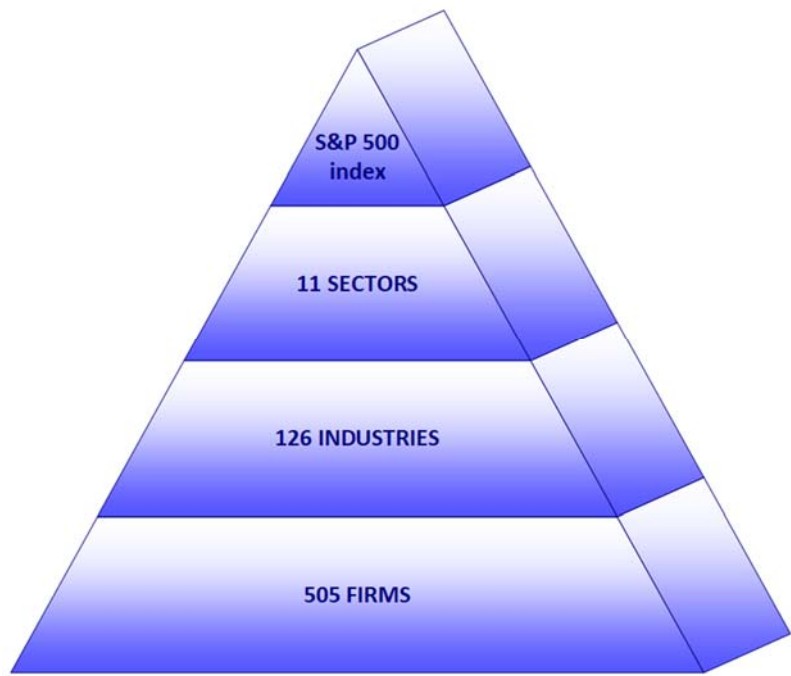

**Figure 1.** Overview of the S&P 500 index, sectors, and industries.

Apart from geopolitical risk, other types of risk are also known to impact the stock return of firms. Examples of other risks include risk from the financial crises (such as recession (BenMim and BenSaïda 2019), banking crises (Miyajima and Yafeh 2007)), risk due to government policy and regulatory decisions (Grout and Zalewska 2006; Lamdin 2001; Jeon et al. 2020), risk due to natural disasters and climate/weather change

(Atems et al. 2020; Bourdeau-Brien and Kryzanowski 2017), risk from political fallouts (Buigut and Kapar 2019), risk as a result of the occurrence of infectious diseases such as COVID-19 and SARs (Mazura et al. 2020; Chen et al. 2007; Bai et al. 2020; Kim et al. 2020), risk due to M&A/joint ventures (Hanvanich and Çavuşgil 2001; Koh and Venkatraman 1991; Park et al. 2002; Dranev et al. 2019; Cuéllar-Fernández et al. 2011), risk due to industrial chemical accidents (Makino 2016), disruption of goods or services in the supply chain (Chen et al. 2019), announcements (Dobija et al. 2012; Hanvanich and Çavuşgil 2001; Jeon et al. 2020), trading of gold and oil market futures (Junttila et al. 2018), trading on the U.S. dollar (the U.S. reserve currency) (Kocaarslan and Soytas 2019), IT infrastructure changes (Wagener et al. 2010), food safety events (Seo et al. 2013), etc.

Caldara and Iacoviello (2018) have developed a geopolitical risk index which measures the fluctuations in geopolitical risk over a given period of time, and we intend to further explore geopolitical risk using this index in order to forecast the impact of this specific risk (GPR) on stock returns of the S&P 500 index. To identify geopolitical risk on the stock prices, we employ an event study which is a commonly used econometric methodology used to estimate the impact of such events (MacKinlay 1997) (Campbell et al. 2012). Obviously, some other methodologies such as structural break and conditional correlation may have been employed.

In our study, we investigate the impact of geopolitical risk on the market returns of the Information Technology, Communication Services, and Consumer Staples sectors of the S&P 500 index, as well as compare the GPR impact between these sector returns. These three sectors were randomly chosen due to the heavy computational work involved in the analysis.

To build on our theory, we map the research domains covered by extant literature to corresponding firm sectors presented in MSCI (2020). To do this, we follow the GICS® (Global Industry Classification Standard) methodology and guidelines presented by MSCI (2020) (e.g., due to the definition of its principal business activity, etc.) to associate research domains covered in extant literature to firm sectors categorized in MSCI (2020). After identifying the firm sector involved in extant research, we investigate how different types of risks affect the stock returns of these identified sectors from extant research. For example, we consider extant research on stocks such as food stocks, cigarette/tobacco stocks, etc., to be associated with stocks in the Consumer Staples sector of the S&P 500 index; we consider extant research on stocks such as entertainment stocks, telecommunication operators' stocks, social media firm stocks (such as Twitter, Facebook, etc.), to be associated with stocks in the Communication Services sector; and consider extant research on stocks such as software stocks, IT stocks, etc., to be associated with the Information Technology sector, in accordance to the guidelines from MSCI (2020).

From the extant literature, we derive the following observations: First, Mazura et al. (2020) provide evidence that diseases, such as COVID-19, cause food stocks (Consumer Staples sector) and software stocks (Information Technology sector) on the U.S. stock market to earn high positive returns, while entertainment stocks (Communication Services sector) fall dramatically. Second, after M&A, acquirer firms in the Information Technology sector experience significant positive average abnormal returns in the short-term and negative average abnormal returns in the long-term (Dranev et al. 2019). Third, Information Technology (IT) sector infrastructure changes (such as the upgrade of the information technology electronic trading system at the Deutsche Börse (the German stock exchange)) lead to an increase in price efficiency (Wagener et al. 2010). Fourth, announcements of joint ventures of parent firms in the U.S. Information Technology sector lead to a significant increase in the market value of the stocks for the participating parent firms. Furthermore, there is a significant and positive relationship between intended and realized joint venture strategies (Koh and Venkatraman 1991). Fifth, the telecommunication operators' (telcos) market (Communication services sector) in China has a negative evaluation of the introduction of 5G technology due to its immaturity and uncertainty. Furthermore, a firm's 5G activities also decrease that firm's value, with this effect having more significance than government policy

announcements (Jeon et al. 2020). Sixth, M&A activities of telecommunications sector firms (Communication Services sector) bring negative news to the market, leading to negative market returns. Furthermore, a cross-border (international) M&A deal is the main driver of the negative market reaction, rather than a domestic M&A deal (Park et al. 2002). Seventh, when acting as a source of sentimental data, Twitter (a Communication Services sector firm) has a statistically strong association with stock prices in the Communication Services sector and that prediction markets manage to effectively pool decentralized information better than alternative sources (Teti et al. 2019). Eighth, food safety events negatively impact food-related firms (Consumer Staples sector), particularly in the first days following the outbreak of events. The effects diminish after approximately two working months after the event and turn to positive after almost one year on from food safety events (Seo et al. 2013). Ninth, according to Atems et al. (2020), a natural weather phenomenon (such as the El Nino-Southern Oscillation (ENSO)) causes a majority of the U.S. food and agricultural stocks (Consumer Staples sector) to have positive and significant returns, resulting in a minority of the stock returns not being significantly different from zero. These effects are, however, short-lived, generally becoming statistically indistinguishable from 3 to 6 months after the shock.

The scope of this study is being confined to the U.S. S&P 500 index because within the last century, the U.S. has been a leading economy in the global market. At the same time, the U.S. has been an active participant in geopolitical events which, as we can infer from Caldara and Iacoviello (2018), should expose its stock market to geopolitical risk (and is also the subject of our investigation). As a leading global economy, the U.S. has Fortune500 companies such as Amazon, Alphabet, Apple Inc., and Microsoft, which have tremendously contributed to the development of technology through innovation and secured a place in history as the first U.S. companies to hit the $1 trillion capitalization value (Berne 2020).

## 2. Literature Review

The Geopolitical risk index is the outcome of a study by Caldara and Iacoviello (2018) aimed at quantifying and creating an index of such (geopolitical) events. By counting the mentions of geopolitical risk in newspapers since 1900, a geopolitical risk index is created through an empirical approach. The study focusses on creating a link between geopolitical risk and their ability to suppress economic activity. It shows that the geopolitical risk index is correlated with a decrease in both economic activity and U.S. stock returns, due to fear of retaliation. According to Caldara and Iacoviello (2018), geopolitical events are a source of risk in the markets. In particular, such events impact different asset classes in different ways, and equities, being an asset class by themselves, are impacted by geopolitical risk.

Amiti et al. (2019) study the correlation between tariffs and changes in consumer prices. This study claims that the revenue captured from U.S. tariffs does not cover the losses incurred by the added cost of consumer imports. The researchers conclude that the global value chains will need to be reorganized if the current tariff policy continues, thus creating further added costs for firms with active investments in the U.S. and China. Qiu et al. (2019) focus on the current trade war based on theories such as imperfect competition, increasing returns, terms of trade arguments, and political economy arguments. The study concludes that protectionism seems to enable specific industries to earn higher returns than the opportunity costs of the resources they possess. However, the benefits of protectionism are diluted by the added complexities that come with such policies, thereby making free trade the path of least resistance. Finally, the study claims that the existing literature on trade cannot fully explain the U.S.–China trade war. While the previous research has focused on the impact of geopolitical events from a theoretical perspective, Huang et al. (2018) focus on the link between trade policies and firm value. The authors emphasize the impact that trade war announcements have on the U.S. and Chinese markets. The study finds a significant correlation between trade policy announcements and stock prices. Every 10 per cent increase in a firm's sales to China produced 0.8 per cent lower average returns.

Chong and Li (2019) compare the ongoing trade war with other historical trade conflicts aiming to reveal the major cause(s) leading to such situations. The authors study the impact on trade by considering the effect on exchange rate between U.S. dollar and Chinese Yuan of historical conflicts of a similar nature. Based on the search for similar events in history, the study concludes that the impact of trade volume decrease between the U.S. and China seems to be over-estimated.

Engle (2019) argues that, while global volatility has been generally low since the end of the global financial crisis, occasional hikes have been observed (such as during U.S. President Donald Trump's election and the Brexit vote), and that volatility across assets, asset classes, industries, and countries tend to be correlated. In addition, he observed that volatility is more predictable, unlike forecasting asset returns, with the ARCH and GARCH models showing that volatility shocks are persistent. Beaulieu et al. (2005) examine the impact of political risks on the volatility of stock returns in Canada. They employ a modified bivariate GARCH model to assess this relationship since this model explicitly measures time-varying financial returns features such as varying volatility and volatility clustering. It also helps in estimating a conditional risk premium and in assessing whether it depends on their proxy for political risk. The authors obtain a sample of 82 political news items (from the U.S.-based Wall Street Journal and The Toronto Globe and Mail) likely to affect the perception of political risk associated with Quebec's independence over the period between 1990 and 1996. Their result shows the relevance and extent to which political news about the possible separation of Quebec (Canada's only mainly French-speaking province out of its 13 provinces and territories) from Canada has on the volatility of stock returns. Furthermore, they show that stock return volatility varies with the degree of a firm's exposure to political risks (political risks affects stock return volatility of domestic firms, but not of international firms) and that investors do not require a risk premium because it can be diversified in a way that does not affect investors' required returns.

Balcilar et al. (2018) research the effects that geopolitical uncertainty have on return and volatility dynamics in Brazil, Russia, India China and South Africa (BRICS) stock markets using nonparametric causality-in-quantiles tests for all BRICS countries. They analyze monthly data on geopolitical risk from the recent work of Caldara and Iacoviello (2018), while the data on the GPR indices are obtained from Iacoviello (2020). They use monthly stock market indices for the BRICS countries and daily stock market returns to calculate the realized volatility estimates for each month. Their results show that the effect of geopolitical risk is heterogeneous across the BRICS stock markets, suggesting that news regarding geopolitical tensions do not uniformly affect return dynamics in these markets. Generally, it is found that news on geopolitical tensions impacts volatility measures, but not the volatility returns, indicating that GPR is a vector of bad volatility in the BRICS markets. They argue that while Russia is seen to have the greatest GPR risk exposure among the BRICS countries in terms of returns and volatility, India is observed to be the most resilient, with geopolitical shocks being undermined through a strong financial apparatus and an open economy. Altogether, the findings suggest that GPRs may be transmitted via volatility interactions across the BRICS markets, with Russia and China acting as the major transmitters of volatility shocks partially driven by geopolitical uncertainties (Balcilar et al. 2018).

Antonakakis et al. (2017) used monthly data from the U.S.-based West Texas Intermediate (WTI) crude oil index and the S&P 500 stock index for a period spanning over a century (1899–2016) to investigate the impact and extent of geopolitical risk on oil-stock covariance, their returns, and their variances, using Caldara and Iacoviello's (2018) GPR index. The authors show that geopolitical risk triggers a negative effect, mainly on oil returns and volatility, and to a lesser extent on the covariance between the stocks and oil markets. Chkili et al. (2014) examine the dynamic relationship between the U.S. stock market and two international crude oil (WT Oil and Brent Oil) markets using a DCC-FIAPARCH model that measures the time-varying properties of conditional return and volatility of both markets as well as the dynamic correlation in the period from 1987–2013. The results show

a strong dynamic conditional correlation (DCC) between these two markets and that the DCC is seen to be impacted by economic and geopolitical events. They surmise that long memory and asymmetric behavior both characterize the conditional volatility of the oil and stock market returns. Finally, they advise investors in the U.S. market to invest more in stocks than oil assets in order to reduce their risk portfolio.

Mazura et al. (2020) investigate the performance of the U.S. stock market during the crash that occurred in March 2020 as a result of COVID-19. Their results provide evidence that the natural gas, food, healthcare, and software stocks earn high positive returns, whereas equity values in petroleum, real estate, entertainment, and hospitality sectors fall dramatically. Furthermore, the stocks that lose exhibit extreme asymmetric volatility that correlates negatively with stock returns. Teti et al. (2019) explore the use of Twitter (Communication Services sector firm) to invest in or verify the relationship with the stock prices in the U.S. communication technology industry. The results show that Twitter (as a source of sentimental data) has a statistically strong association with stock prices in the Communication Services sector and that prediction markets manage to effectively pool decentralized information better than alternative sources. Further results indicate a higher association between the stock prices of companies with high social media coverage than that with low coverage.

Dranev et al. (2019) investigate the post-acquisition performance of the acquirer firms in both the Information Technology and financial sectors (fintech) measured by abnormal returns after recent mergers and acquisitions. They discover significant positive average abnormal returns after the acquisition of fintech companies in the short-term and negative average abnormal returns in the long-term using event studies, which is consistent with some prior studies on technology M&A.

Atems et al. (2020) examine the response of twelve U.S. food and agricultural stock returns to El Nino-Southern Oscillation (ENSO) (a naturally occurring weather phenomenon that involves fluctuations in winds and ocean surface temperatures in the central and east-central equatorial Pacific Ocean) shocks using a recursive VAR model. Their results indicate that, for seven of the stock returns, an ENSO shock has positive and significant effects, while five of the 12 stock returns are not significantly different from zero. However, the effects are short-lived, generally becoming statistically indistinguishable from three to six months after the shock. Variance decomposition analyses show that ENSO shocks have little explanatory power for fluctuations in U.S. agricultural stock returns. They also find that, historically, movements in U.S. agricultural stock returns have been driven by other shocks, rather than ENSO shocks.

Wagener et al. (2010) use an event study analysis and show that IT infrastructure changes such as the upgrade of the information technology electronic trading system at the Deutsche Börse (the German stock exchange) led to an increase in price efficiency, measured in terms of the price gaps between the observed future prices and their theoretical values in the cash market. Seo et al. (2013) use event studies to investigate how food safety events (e.g., the 2002 *E. coli* bacteria outbreak) impact the value of food-related firms, based on a conceived theoretical model relating food safety events and firm value. They observe that food safety events have negative impacts on food-related firms, particularly noting that the daily abnormal returns (ARs) are significantly negative on the first day ($t_1$) and the second day ($t_2$) following the outbreak of events. Cumulative abnormal returns (CARs) are found to be significantly negative and diminished after approximately two working months after the event. Furthermore, the negative CARs turn to positive values after almost one year from food safety events. Additionally, from the results obtained, firm-specific factors such as past history, firm size, and the amplifying effect of media messages on poor firm values are significant factors influencing changes in stock returns due to food safety event outbreaks.

Jeon et al. (2020) examine the impact of the government's 5G policy announcements on telecommunication operators' (telcos) firm value in China, where the government has great control. They find that government policy announcements, in general, negatively affect

telecommunication operators' stock return, and when the government announces policies with a higher level of interference, the decline in firm value is more pronounced. Further results show that the firms' 5G activities also decrease their firm value, and this effect is more significant than government policy announcements. These results imply that the market has a negative evaluation of the introduction of 5G technology due to its immaturity and uncertainty. Koh and Venkatraman (1991) analyze the impact of joint venture formation strategies on the market value of parent firms in the information technology sector using the event study methodology and conclude that announcements of joint ventures lead to a significant increase in the market value of the stocks for the participating parent firms. Further analysis of the impact of intended and realized joint venture strategies using a subsample also reveal a significant and positive relationship. Park et al. (2002) evaluate how telecommunications firms react to mergers and acquisitions (M&As). Their findings showed that M&A activities for these sector firms bring negative news to the market, leading to negative market returns. Furthermore, a cross-border (international) M&A deal is seen to be the main driver of the negative market reaction, rather than a domestic M&A deal. Baur and Smales (2020) investigate the relationship between geopolitical risk and asset prices of precious metals such as gold, silver, platinum, and palladium, then compare these with industrial metals such as copper. Their results show that precious metals are hedges to geopolitical threats as a form of geopolitical risk, and not to geopolitical acts. Further results show that stocks and bonds respond negatively to geopolitical risk and geopolitical threats. Another result shows that, for extreme geopolitical risks, only gold and silver consistently portray a safe haven for investment, and that a way of lowering the impact of geopolitical risk is to hold precious metals within a diversified portfolio.

Fama (1970) introduces the efficient market hypothesis, whereby the price of a stock at a given time captures and reflects the accumulation of all publicly available information. Furthermore, he argues that the value of a firm is quantified by its stock price. The stock market is one instance where the efficient market hypothesis can be applied. To test the efficient market hypothesis in a selected subset of a stock market, Fama (1991) recommends the use of an event study. Such studies can be used to quantify the effect(s) of a certain event (such as the impact of a geopolitical event) towards firms in a particular economic sector. According to Campbell et al. (2012), the market model has been the dominant model choice for event studies since 1970, and in the market model, there is a built-in assumption that the expected returns of stocks are linearly related to the returns of the market from which the specific stock is part of Bowman (1983) presents event study research as a methodology that originates not only from announcements made by firms, but also announcements made from outside of firms (e.g., an accounting standard body such as the FASB) or from other general events/happenings such as an oil embargo. This paper presents a framework for designing event studies, differentiates them by type, and carries out more exploratory analysis on event studies. The five steps of event studies as well as the identification of alternative techniques, to provide a basis for understanding and conducting event studies are discussed.

A comprehensive study of event studies is presented by MacKinlay (1997). In his paper, MacKinlay (1997) reviews and summarizes event study methods and discusses a possible procedure for conducting an event study. He presents statistical choices for modelling the normal return such as the Constant Mean Return Model, the Market Model, and the Factor Model. He also presents economic models that provide more constrained normal returns such as the Capital Asset Pricing Model (CAPM), and the Arbitrage Pricing Theory (APT). Binder (1998) reviews developments in the event study methodology beginning with Fama et al. (1969). Some of these developments include hypothesis testing, the use of different benchmarks for the normal rate of return, the power of event studies in different applications, and the modelling of abnormal returns as coefficients in a (multivariate) regression framework. Binder (1998) also researches frequently encountered statistical problems in event studies and their solutions. Kwok

and Brooks (1990) examine the event study methodology used by Brown and Warner in the foreign exchange market, by comparing the performance of four alternative abnormal return models (the random walk model, the mean adjusted returns model, the market-adjusted returns model, and the market model) under different experimental conditions. Some of their findings agree with those of Brown and Warner, while some do not. Further results show that a careful design of the methodology may improve its sensitivity and demonstrate that some of the findings of Brown and Warner are not generalizable to the foreign exchange market.

Barnhill and Maxwell (2002) go beyond the current portfolio risk estimation methodologies that calculates market and credit risk in separate analyses, to introduce a methodology that combines these risk measures into one overall portfolio risk assessment to assess the correlated market and credit risk. The risk assessment methodology applies to a single bond demonstrates that out of the four important risk factors (interest rate, spread, credit, and FX (foreign exchange) risk), the most important for non-investment grade bonds is credit risk. Klein and Rosenfeld (1987) use simulation techniques as well as an actual event to examine the reliability of four separate return-generating models to test for possible biases in the specification of abnormal returns for the bull (periods of growth) and bear (periods of decline) markets. They find minor differences only among the models over the 2-day announcement period. Their results further show that both the pre-event and post-event CARs (Cumulative Abnormal Returns) are significantly positive (negative) during bull (bear) markets for those models that do not adjust for market trends, such as the mean-adjusted and raw-market returns. Szegö (2002) discusses some classical measures of risk such as the mean, linear correlation coefficient, and Value at Risk (VaR). The author presents measures to measure risks such as Conditional Value at Risk (CVaR) and Expected Regret (ER) and exposes the limitation of the VaR in measuring risk.

In his paper, Agmon (1973) measures and describes the share-price fluctuations associated uniquely with their country of origin and estimates the country factor in share-price movements in the equity markets of the United Kingdom, Germany, and Japan. The findings show that although movements of share prices in the equity markets of the United Kingdom, Germany, and Japan are related to price changes in the U.S. market index, once the U.S. factor is removed, the non-U.S. country indices are virtually independent of each other, as there is another residual factor affecting share-price fluctuations in these three markets.

Cowan and Sergeant (1996) examine the effects of thin trading on the specification of event study tests using simulations of upper and lower tail tests with and without variance increases on the event date across levels of the trading volume. The authors note that the traditional Patell standardized test is poorly specified for thinly traded stocks, so the best way to replace the standardized test depends on the conditions of the study. Accordingly, they observe that the rank test presented the best specification and power for general use if the return variance is unlikely to increase on the event date. Additionally, they report that the rank statistic is incorrectly specified in lower-tailed tests regardless of volume and in upper-tail tests for several samples, while they recommend that the standardized cross-sectional statistic should be preferred for upper-tailed tests and the generalized sign statistic for lower-tailed tests. Vennet (1996) analyzes a sample of 492 takeovers to examine the performance effects of acquisitions and mergers between EC credit institutions over the period 1988–1993. A performance analysis is performed to determine whether or not bank mergers entail real economic gains. The results indicate that domestic mergers among equal-sized partners significantly increase the performance of the merged banks. Vennet (1996) further reports that an improvement of cost efficiency is also found in cross-border acquisitions, while, conversely, domestic takeovers are found to be predominantly influenced by defensive and managerial motives such as size maximization.

The properties of daily stock returns and the ways in which some particular characteristics of these data affect event study methodologies are investigated by Brown and Warner

(1985). Using the results from simulations, they conclude that methodologies based on the OLS market model and using standard parametric tests are well-specified under a variety of conditions. Further results indicate a striking similarity between the empirical power of the event study procedures and the theoretical power. Sykulski (2014) presents the concept of geopolitical risk as consisting of possible threats resulting from international competition between states for access to and use of natural resources, the expansion of their zones of political and economic influence, as well as competition for control of strategic areas. The author refers to international relations as cross-border relations between political entities and argues that there is a lack of central authority (a regulator) capable of managing the political life of international entities in an organized manner.

### 3. Data

Based on existing research, such as Campbell et al. (2012), the effect of an event is directly/immediately reflected on the stock prices (Campbell et al. 2012, p. 149). Consequently, the effect of geopolitical events has a direct economic impact on stock prices. In our data analysis design, we use the data series consisting of historical stock prices of firms in the Information Technology, Communication Services, and Consumer Staples sectors on the S&P 500 index.

Due to the large number of regressions needed and the heavy computational effort involved in calculating the expected stock returns for the 505 firms from all 11 sectors of the S&P 500 index, and over a large number of geopolitical events, we choose to randomly sample three economic sectors as follows: Information Technology, Communication Services, and Consumer Staples. This resulted in the analysis of 131 firms belonging to these three sectors.

Since not all firms are listed in the stock exchange since 1962, for each event, a subset of firms is available to be averaged in order to calculate the AR. For example, one of the firms under study was The Hershey Company (ticker: HSY) which belongs in the Consumer Staples sector of the S&P 500 index. Hershey has been listed in the stock market since 17 March 1980, according to the Yahoo! Finance historical database (Yahoo! Finance 2020). Therefore, for this company, we have not been able to perform the event study steps for the events of interest preceding 17 March 1980.

The firms' daily return historical data are collected for each of the sampled firms from Yahoo! Finance (Yahoo! Finance 2020). This source is chosen to minimize the effort of creating primary data while leveraging on the already existing free and up-to-date database of Yahoo! Finance. Moreover, this database includes a vast amount of historical information for indexes as well as individual stocks which would be of key importance for this study. Since the sampled companies in the S&P 500 from the selected economic sectors have available stock price data starting in 1962 (at the earliest), the choice is made to investigate only the impact of geopolitical risk on the U.S. equities between 1962 and 2020.

After successfully collecting the stock and index price data, the focus of data collection shifts to the GPR index historic values. Based on the study carried out by Caldara and Iacoviello on the GPR index (Caldara and Iacoviello 2018), the data are collected from the outcome of that study. On their website, they have been continuously providing an updated version of the GPR index on a monthly as well as a daily basis. Moreover, they have been creating charts, illustrating the actual events associated with each GPR spike (Iacoviello 2020). Within the scope of this study, the selection of geopolitical events under study is limited to the named and highlighted events found in the historical GPR index charts, which in total accounted for 73 events, ranging from 1900 until 2020. However, due to the availability of individual stock data from 1962, the focus of this study is the events starting from 1962, and as a result, we ignore previous events before 1962. This results in 58 events, starting from 1962.

In line with the second step from Campbell et al. (2012), we define, as events of interest, the 58 geopolitical events between 1962 until 2020 from Caldara and Iacoviello's historical GPR chart (Iacoviello 2020). Each event is connected to a specific date when the

GPR index spiked. Descriptive Statistics for the daily-returns of the sampled firms for the event "Moscow theater hostage crisis" for the (−10,10) event window are given in the Appendix A (see Table A1). A list of geopolitical events from 1962–2020 is given in the Appendix A (Table A2). The average abnormal returns and cumulative average abnormal returns for the three sectors for the same event and event window can be found in Table A3 of Appendix A.

To perform the event study analysis in our research, we followed the seven systematic steps as presented by Campbell et al. (2012).

According to Campbell et al. (2012), the first step is captured as follows: "The initial step when performing an event study is to define the events of interest and the timespan in which the security's price will be examined, also known as the event window".

For the event study timeline, the framework shown in Figure 2 is chosen (MacKinlay 1997).

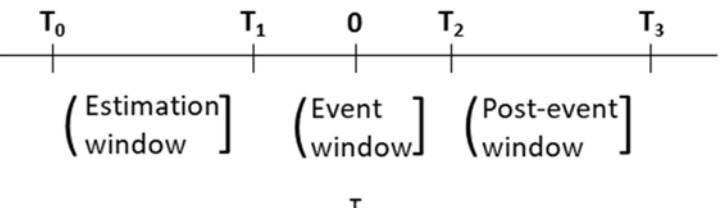

**Figure 2.** Event study framework timeline.

For each of the events, we define $T_0$ to denote the start of the estimation window, and $T_1$ as the end of the estimation window. The duration of the estimation window is then represented as $(T_0, T_1)$. Then, $T_{1+1}$ is defined as the start of the event window, "0" is set as the event date that the GPR index spiked, and $T_2$ is set as the end of the event window. The duration of the event window is then represented as $(T_{1+1}, T_2)$. For this study, four different event windows are chosen to keep the scope to a manageable level. For the first scenario (−3,3), $T_{1+1}$ is set to 3 days before the event date, and $T_2$ is set to 3 days after the event in order to capture short-term reactions before and after the actual event. Then, a (0,5) event window is selected to capture only post-event fluctuations in the share prices. The (0,1) event window is chosen to capture the immediate impact of the event after its occurrence. Finally, a (−10,10) window is chosen to cover the case where early information leakages have started to adjust the market price before the actual event. Descriptive statistics for the S&P 500 index, Information Technology, Communication Services, and Consumer Staples sectors for estimation period and event windows are given in Table 1.

To be more accurate, an event should have occurred 253 working days after the day a firm was listed in the stock market for us to be able to perform an event study analysis and calculate the AR for that firm. This is because we have chosen an estimation window of 253 days to capture a full calendar year before a particular event, as described in more detail. Table A4 in Appendix A captures the number of events that are possible to analyze for all 131 firms under investigation in the Information Technology, Communication Services, and Consumer Staples sectors.

**Table 1.** Descriptive statistics for the daily returns (multiplied by 100) of the S&P 500 index, Information Technology, Communication Services, and Consumer Staples sectors for the (−10,10) event window and the "Moscow theater hostage crisis" event.

|  | Period | Obs. | Mean | Std. Dev. | Maximum | Minimum | Skewness | Kurtosis |
|---|---|---|---|---|---|---|---|---|
| S&P 500 Index | $T_0–T_1$ | 253 | 0.030 | 0.960 | 4.959 | −3.286 | −0.064 | 4.287 |
|  | $T_{1+1}–0$ | 10 | −0.012 | 0.581 | 0.739 | −1.089 | −0.409 | −0.952 |
|  | $0–T_2$ | 10 | −0.351 | 1.621 | 1.876 | −2.978 | −0.381 | −0.837 |
|  | $T_{1+1}–T_2$ | 21 | −0.216 | 1.209 | 1.876 | −2.978 | −0.674 | 0.780 |
| Information Technology | $T_0–T_1$ | 253 | −0.106 | 2.574 | 9.219 | −5.068 | 0.461 | 0.102 |
|  | $T_{1+1}–0$ | 10 | 1.633 | 3.924 | 6.822 | −5.879 | −0.428 | −0.618 |
|  | $0–T_2$ | 10 | 1.548 | 1.867 | 4.611 | −0.167 | 0.575 | −1.863 |
|  | $T_{1+1}–T_2$ | 21 | 1.374 | 3.151 | 6.822 | −5.879 | −0.213 | −0.169 |
| Communication Services | $T_0–T_1$ | 253 | −0.013 | 2.135 | 7.144 | −7.358 | 0.173 | 0.851 |
|  | $T_{1+1}–0$ | 10 | 0.138 | 1.951 | 3.186 | −2.207 | 0.490 | −1.427 |
|  | $0–T_2$ | 10 | 1.029 | 1.348 | 3.814 | −0.460 | 0.959 | −0.117 |
|  | $T_{1+1}–T_2$ | 21 | 0.534 | 1.707 | 3.814 | −2.207 | 0.328 | −0.824 |
| Consumer staple | $T_0–T_1$ | 253 | 0.027 | 1.006 | 3.990 | −4.032 | −0.061 | 1.625 |
|  | $T_{1+1}–0$ | 10 | 0.023 | 1.298 | 1.987 | −1.725 | −0.052 | −1.635 |
|  | $0–T_2$ | 10 | −0.107 | 0.662 | −0.597 | −1.943 | −2.267 | 6.531 |
|  | $T_{1+1}–T_2$ | 21 | 0.004 | 1.028 | 1.987 | −1.943 | −0.294 | 0.293 |

## 4. Model

In this paper, we want to capture the effect of an event or GPR on stock return. If an event has an impact on the stock return, we may expect that the return of the stock will deviate from its normal return. Several methods such as the factor model, Capital Asset Pricing model (CAPM), Arbitrage Pricing Model (APT), constant mean return model, and market model could be used to measure the impact of GPR in a time series framework. In this paper, we study the impact of GPR on stock return, hence we may employ CAPM, APT, and the market model. The reason we chose this approach rather than other options (such as the constant mean return model and the asset pricing model) is because the market model has higher validity (Campbell et al. 2012). A vivid discussion on the model choice is to be found in (MacKinlay 1997). The market model according to MacKinlay (1997) can be written as follows:

$$R_{it} = \alpha_i + \beta_i R_{mt} + e_{it} \tag{1}$$

where $E(e_{it}) = 0$ and $Var(e_{it}) = \sigma_{ei}^2$. The $R_{it}$ denotes the return of a firm $i$ at time $t$ and $R_{mt}$ represents the return of the market ($m$) and time $t$. The expected value of (1) is:

$$E(R_{it}) = \hat{\alpha}_i + \hat{\beta} R_{mt} \tag{2}$$

which is considered as a normal return of the stock $i$. The effect of an event can be reflected in stock return deviating from the normal return which we call abnormal return (MacKinlay 1997). Following Equation (2) above (MacKinlay 1997), which is in accordance with the third step from Campbell et al. (2012), we calculate the daily expected returns for each company based on the market model.

$$AR_{i\tau} = R_{i\tau} - \hat{\alpha}_i + \hat{\beta} R_{m\tau}. \tag{3}$$

The abnormal return $AR_{i\tau}$ is the deviation from or disturbance term of the market model. The $\tau \in [T_{1+1}, T_2]$ which is an out of sample or the event window. Condition of the event window and under the null hypothesis $AR_{i\tau} \sim (0, \sigma(AR_{i\tau}))$ where:

$$\sigma(AR_{i\tau}) = \sigma_{ei}^2 + \frac{1}{L_1}\left[1 + \frac{(R_{m\tau} - \hat{\mu}_m)^2}{\sigma_m^2}\right] \tag{4}$$

The $L_1$ is the length of the estimated window, $\hat{\mu}_m = \frac{1}{L_1} \sum_{\tau=T_0+1}^{T_i} R_{m\tau}$ and $Var(R_{mt}) = \sigma_m^2$.
The second part of the right-hand side approaches zero as $L_1$ increases. An observation of the abnormal return in a window under the null hypothesis is $AR_{i\tau} \sim N(0, \sigma(AR_{i\tau}))$ (MacKinlay 1997). In order to draw inferences for the event of interest, the abnormal return must be aggregated. We can aggregate over time and across firms. The sample cumulative abnormal return $CAR_i(\tau_1, \tau_2)$ over time is:

$$CAR_i(\tau_1, \tau_2) = \sum_{\tau=\tau_1<T_1}^{\tau_2 \le T_2} AR_{i\tau} \tag{5}$$

Asymptotically, the variance of cumulative abnormal return $CAR_i(\tau_1, \tau_2)$ is:

$$\sigma_i^2(\tau_1, \tau_2) = (\tau_2 - \tau_1 + 1)\sigma_{ei}^2 \tag{6}$$

This estimator of variance for the large sample can be used for reasonable values of $L_1$. For a small sample, the variance of cumulative abnormal return should be adjusted for estimation error. Under the null hypothesis, the distribution of the cumulative abnormal return is:

$$CAR_i(\tau_1, \tau_2) \sim N\left(0, \sigma_i^2(\tau_1, \tau_2)\right). \tag{7}$$

For $N$ events, the sample average abnormal return for an individual firm $i$ and period $\tau$ is:

$$AAR_t = \overline{AR_\tau} = \frac{1}{N} \sum_{i=1}^{N} AR_{i\tau} \tag{8}$$

Additionally, for large $L_1$, the corresponding variance is:

$$VAR\left(\overline{AR_\tau}\right) = \frac{1}{N^2} \sum_{i}^{N} \sigma_{ei}^2 \tag{9}$$

The cumulative average abnormal return for $n$ days within the event window $\tau$ is:

$$CAAR_e = \frac{1}{n} \sum_{\tau=\tau_1<T_1}^{\tau_2 \le T_2} AAR_\tau \tag{10}$$

$$AR\left(\overline{CAR}(\tau_1, \tau_2)\right) = \sum_{\tau=\tau_1<T_1}^{\tau_2 \le T_2} VAR(\overline{AR_\tau}). \tag{11}$$

Alternatively, and equivalently, we can form CAARs firm by firm and the aggregate through time (MacKinlay 1997). Finally, after iterating through the 58 geopolitical events identified earlier in the study (from the spikes on the GPR index), the events that resulted in a statistically significant CAAR are recorded and passed to the final stage of the analysis. In the final stage, we calculate the historical average CAAR as follows:

$$HistoricalACAAR = \frac{\sum_{0}^{Ne} CAAR_0}{N_e} \tag{12}$$

where $N_e$ is the Number of events with statistically significant CAAR. Finally, the last step according to Campbell et al. (2012) is "*Interpretation and insights from the results are made. Additional analyses may be necessary to single out relevant factors that affect the share prices*". This step is fulfilled in Section 5 (Empirical findings and Analysis).

*4.1. Estimation Technique*

The fourth step would be to set the estimation window according to Campbell et al. (2012) as follows: "After selecting the model, the parameters must be estimated by using data from before the event period, also known as the estimation window".

The estimation window is the time period before the event window starts, during which data is gathered to estimate the expected returns during the event window. For our study, an estimation window of 253 days before the event window starts $(T_1)$ is chosen. This choice was made because there are, on average, 253 trading days in the year on which the U.S. stock market is open, and we want to capture a full-year ahead of an event in our analysis. By doing this, we guarantee that we have collected enough data for our market model to produce an accurate estimation of expected returns. To avoid any potential biases that the events of interest could introduce during the data collection process within the estimation window, we chose not to overlap the estimation window with the event window. From the framework presented in Figure 2, the duration between $T_0$ and $T_1$ was set to 253 days.

*4.2. Testing Procedure*

According to Campbell et al. (2012), the fifth step is as follows: "*Once the estimation procedure was completed and the parameters estimated, the abnormal returns could be calculated. The framework in which the abnormal returns are tested must be designed and null hypothesis should be defined*".

To prove a causal relationship (or a covariation) between a geopolitical event and the stock returns, the results must be statistically significant (Ghauri and Grønhaug 2010, p. 83). To verify if the AAR for a given geopolitical event is statistically significant, a *t*-test according to Müller (2020) is performed as follows:

$$AARtvalue_e = \sqrt{N}\frac{AAR_e}{StdDev_{AAR}} \tag{13}$$

where:

$AARtvalue_e$ = Value of *t*-test for the AAR for event *e*;
$StdDev_{AAR}$ = Standard deviation of the AAR values within the estimation window;
$N$ = the number of AR's (from different firms) averaged to create the AAR for that date.

If the *t*-test of an AAR value results in an absolute value greater than 1.96 ($|AARtvalue_e| > 1.96$), it implies that the specific AAR is statistically significant at 0.05. To verify if the CAAR for a given geopolitical event is statistically significant, a *t*-test according to Müller (2020) is performed as follows:

$$CAARtvalue_e = \frac{CAAR_e}{StdDev_{AAR} \times \sqrt{Number\ of\ days\ in\ the\ event\ window}} \tag{14}$$

where:

$CAARtvalue_e$ = Value of *t*-test for the CAAR for event *e*
$StdDev_{AAR}$ = Standard deviation of the AAR values within the estimation window

If the t-test of a CAAR value results in an absolute value greater than 1.96 ($|CAARtvalue_e| > 1.96$), it implies that the specific CAAR is statistically significant at 0.05. To be able to test the null hypothesis for a given geopolitical event, a criterion was added. This criterion requires the CAAR for a given geopolitical event as well as at least 1 AAR within the event window to be statistically significant at 0.05.

## 5. Empirical Findings and Analysis

Since we consider four event windows and 58 geopolitical events in our study, Table A5 (Appendix A) captures the total number of regressions to be performed for a full S&P 500 index event study analysis. The difference between the possible number of regressions (117,160) and the actual number of regressions (73,140) is because some

of the firms were not publicly listed when certain geopolitical events occurred. Table A6 (Appendix A) depicts both the possible number of regressions (30,392) and the actual number of regressions (17,388) performed within the scope of the study. This difference is due to the same reason as above. Therefore, for this study, more than 17,000 regressions were computed as part of the event study analysis.

In Figure 3, we visualize an example of the abnormal returns for Hershey within a (−3,3) event window for the Moscow theater hostage crisis (a geopolitical event that took place on 23 March 2002). The grey area in Figure 3 represents a part (subsection) of the estimation window. Within the estimation window, the necessary input (intercept—$\alpha_i$ , slope—$\beta_i$) from the market model (Equation (1)) is derived and based on that, the expected daily returns (including within the event window) for the firm are forecasted (represented by the dotted line with circle markers). The line with square markers in the graph (Figure 3) depicts the actual returns of the firm as obtained from the website of Yahoo! Finance (Yahoo! Finance 2020), while the line with triangle markers represents the difference between the actual and expected daily returns (also known as the abnormal returns), which is derived from Equation (3).

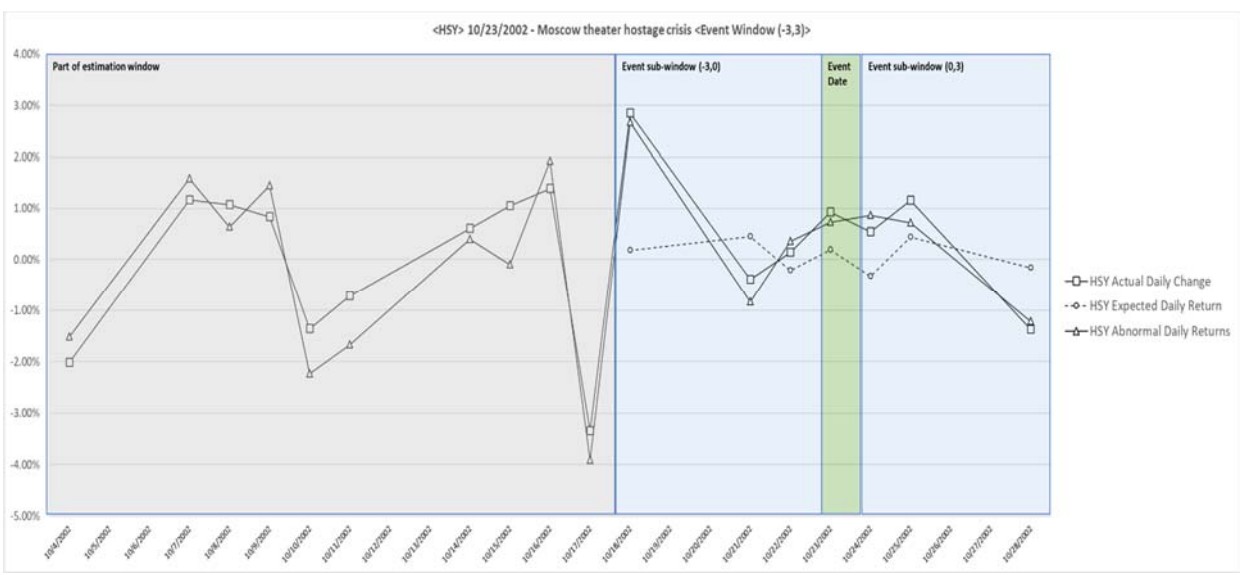

**Figure 3.** HSY's daily abnormal returns to the Moscow theater hostage crisis: Event window (−3,3).

For illustration purposes, we divide the event window (−3,3) into two event sub-windows ((−3,0) and (0,3)), shown by the two light-blue background sections in Figure 3. The actual occurrence of the geopolitical event (the Moscow theater hostage crisis), also known as the "Event Date", is captured between the two event sub-windows (and represented by the light-green background in Figure 3). After capturing the returns within the event window for this event we calculate the average abnormal returns for each day within the event window (−3,3).

To visualize the abnormal returns (AR) as well as the cumulative abnormal returns (CAR) of individual firms during a geopolitical event and within an event window, we make use of Equations (3) and (5), respectively. An example of AR and CAR results for three firms, with one firm each from the individual sectors under study is presented in Figures A1 and A2, respectively. In this example, we have used Apple (ticker:AAPL) from the Information Technology sector, Disney (ticker:DIS) from the Communications Services sector, and Coca-Cola (ticker:KO) from the Consumer Staples sector.

To calculate the CAAR per geopolitical event for the Information Technology, Communication Services, and Consumer Staples sectors, we average (for each geopolitical event) the statistically significant AR values for all companies within the sectors (Information Technology, Communication Services, and Consumer Staples). The final results consist

of the calculation and presentation of the Historical Average CAAR (using Equation (12)) for the Information Technology, Communication Services, and Consumer Staples sectors. This is achieved by averaging the individual geopolitical event's Information Technology, Communication Services, and Consumer Staples CAAR values for different events.

Table 2 below depicts the final results and descriptive statistics for this study. The number of statistically significant events per sector and event window is presented, as well as the average, min, max, and standard deviation of the CAAR values for the Information Technology, Communication Services, and Consumer Staples sectors.

**Table 2.** Summary of Historical Sector cumulative and average CAAR results for all events windows.

| Sectors | (−3,3) | | | (0,5) | | | (0,1) | | | (−10,10) | | |
|---|---|---|---|---|---|---|---|---|---|---|---|---|
| | Information Technology | Communication Services | Consumer Staples | Information Echnology | Communication Services | Consumer Staples | Information Technology | Communication Services | Consumer Staples | Information Technology | Communication Services | Consumer Staples |
| Number of Statistically significant Events | 4 | 1 | 7 | 2 | 3 | 6 | 5 | 2 | 10 | 5 | 6 | 5 |
| Historical Sector Average CAAR | −7.75% | 2.94% | −1.16% | −5.14% | −0.49% | 0.90% | −1.30% | 2.62% | −0.71% | 4.31% | 0.97% | −1.30% |
| Historical Sector CAAR Standard Deviation | 2.07% | 0.00% | 3.89% | 0.00% | 5.55% | 2.97% | 5.24% | 1.29% | 1.70% | 15.61% | 12.52% | 6.12% |
| Historical Sector Min CAAR | −10.44% | 2.94% | −4.28% | −5.14% | −3.67% | −3.31% | −7.44% | 1.71% | −2.69% | −12.50% | −21.10% | −7.19% |
| Historical Sector Max CAAR | −5.41% | 2.94% | 4.88% | −5.14 | 5.92% | 4.58% | 4.73% | 3.54% | 2.49% | 19.58% | 13.02% | 6.04% |

The percentage of events with statistically significant CAAR results (per event window) for the Information Technology, Communication Services, and Consumer Staples sectors is presented in Figure A4.

By examining Table 2 and Figure A3, and specifically looking at the Historical Sector Average CAAR results, it becomes clear that there are different responses on the daily returns of stocks belonging to the Information Technology, Communication Services, or the Consumer Staples sectors. The Historical Sector Average CAAR of the Information Technology sector is negative for all event windows, except the (−10,10) event window. The Communication Services sector has positive returns as a result of geopolitical events for all event windows, except the (0,5) event window. Finally, the Consumer Staples sector shows a negative impact on geopolitical risk for all event windows except the (0,5) window, demonstrating a negative correlation to the Communication Services sector.

To understand the distribution of the results, we use the standard deviations of the Historical Average CAARs for the three sectors per event window. It is clear from Table 2 that the standard deviations of all three sectors increase in the (−10,10) event window compared to the other event windows. We also observe that the Information Technology sector has the worst negative impact as a result of geopolitical events compared to the Consumer Staples sector for all event windows, when examining the Historical Min CAAR. Finally, when examining the Historical Max CAAR, the results show that the Information Technology sector has the worst negative impact as a result of geopolitical events compared to the other sectors for the (−3,3) and (0,5) event windows, while for the (0,1) and (−10,10) event windows, it has the most positive impact from geopolitical events.

From the result, we observe the impact of geopolitical risk/events on the three different sectors is non-uniform when considering the Historical Sector Average CAAR across four event windows. This is in line with the findings of Caldara and Iacoviello (2018) and Iacoviello (2020), whose results point to an asymmetric (non-uniform) impact of geopolitical risk between industries—something that we also validate with our results. Furthermore, our results from studying the impact of geopolitical events on the Information Technology sector contradict the results of Mazura et al. (2020), and Dranev et al. (2019). However, we note that their research focused on different kinds of risks (COVID-19 and M&A,

respectively) than the geopolitical risk that we are researching on. Additionally, our results from studying the impact of geopolitical events on the Communication Services sector contradicts the results of Jeon et al. (2020) and Park et al. (2002). However, we note that their research focused on different kinds of risks (introduction of 5G technology and M&A, respectively) than the geopolitical risk that we are researching on. Finally, our results obtained from studying the impact of geopolitical events on the Consumer Staples sector agrees with the results of Seo et al. (2013) and Atems et al. (2020) in the short run. We also note here that their research focused on risks of a different nature (food safety events and a natural weather phenomenon, respectively) than the geopolitical risk that we are researching.

Figure A4 summarizes the percentages of events that generated statistically significant CAAR for the Information Technology, Communication Services, and the Consumer Staples sectors across all four event windows. In particular, we observe that the Consumer Staples sector the highest number of events with statistically significant CAAR for the (−3,3) and (0,1) event windows compared to the other two sectors; at the same time, for the (0,5), and (−10,10) event windows, it has the least number of events with statistically significant CAAR compared to the other two sectors. On the other hand, the Communication Services sector has the highest number of events with statistically significant CAAR for the (0,5), and (−10,10) event windows compared to the other two sectors; at the same time, for the (−3,3) and (0,1) event windows, it has the least number of events with statistically significant CAAR compared to the other two sectors.

Our results have high validity because we ran the event study procedure over multiple event windows ((−3–3), (0,5), (0,1), and (−10,10)) with different interval durations (7 days, 6 days, 2 days, and 21 days, respectively), to capture the abnormal stock returns.

The results of our study are highly reliable for the following reasons: First, we chose an estimation window of 253 days to guarantee that we have collected enough data for the market model to produce an accurate estimation of expected returns. Second, we limited potential bias and improved the reliability and robustness of our results by performing the event study that applies the same parameters and repeats the same procedures on firms in the Information Technology, Communication Services, and Consumer Staples sectors. Third, we were also able to improve the reliability of our results by choosing not to overlap the estimation window with the event window, thereby reducing potential biases that the events of interest could have introduced within the estimation window.

Our results are ethically sound and objective because, during the process of obtaining these results, we collected public data after which we used strict and consistent procedures to analyze the data following the seven systematic steps of the event study analysis as presented by Campbell et al. (2012). Additionally, we prevented ethical issues by clearly defining the period of 1962–2020 as the observed period we consider for geopolitical events in this study.

## 6. Discussion

We began this study intending to explore the impact of geopolitical risk on the stock returns of the Information Technology, Communication Services, and Consumer Staples sector firms belonging to the U.S. S&P 500 index and compare the GPR impact between these sector returns. Geopolitical risk is a result of uncertainty introduced by geopolitical events, and we captured these events using the benchmark GPR index of Caldara and Iacoviello (2018). We have been able to measure this impact using the event study analysis tool (MacKinlay 1997; Campbell et al. 2012) which includes a longitudinal study of events from 1962 to 2020, and an evaluation of the average Historical CAAR results of the Information Technology, Communication Services, and Consumer Staples sectors from the historical stock prices of Information Technology, Communication Services, and Consumer Staples sectors firms belonging to the S&P 500 index. The event study makes use of the efficient market hypothesis which helps in analyzing the impact that geopolitical risk

has on the Information Technology, Communication Services, and the Consumer Staples sectors.

The data for our study are collected from publicly available websites such as Yahoo! Finance (2020) and Slickcharts (2020). We also employed the quantitative empirical analysis as the preferred method in our study to analyze the data and followed the seven systematic steps of the event study methodology to quantitatively measure the operationalized variables (across multiple event windows with different interval durations). The above approaches strengthen the validity, robustness, reliability and ethics as well as the soundness of our results.

Our results are in tandem with the conclusions made by Caldara and Iacoviello (2018) and Iacoviello (2020), stipulating the asymmetric impact of geopolitical risk between the stock returns of different industries, where a sector (which is the area of our investigation) is a collection of industries. Our results also agree with those of Seo et al. (2013) and Atems et al. (2020) in the short run on their research about the Consumer Staples sector. However, our results do contradict the results of Mazura et al. (2020) and Dranev et al. (2019), and those of Jeon et al. (2020) and Park et al. (2002) who conducted research on the Information Technology and Communication Services sectors, respectively.

The results of our study can be generalized to other settings in terms of stock exchanges, firm economic sectors, and geography, as long as they closely resemble the S&P 500 index as a market.

*6.1. Contributions, Implications, and Applications*

In our study, we investigate the effect that geopolitical risk has on the daily abnormal returns of the Information Technology, Communication Services, and Consumer Staples sector firms within the U.S. S&P 500 index. The main contribution of our study is that we use the event study methodology and perform more than 17,000 regressions to provide empirical evidence at sector level (and not at industry level) that geopolitical risk leads to different responses across the Information Technology, Communication Services, and Consumer Staples sectors. This complements the results from Caldara and Iacoviello (2018), who use the VAR model to analyze the impact of geopolitical risk on an industry-level.

The results of our study have some implications. In the short run (for the ($-3$,3), (0,1), and (0,5) event windows), the Information Technology sector appears to have the worst impact from geopolitical events compared to the Communication Services and Consumer Staples sectors. However, in the long run (the ($-10$,10) event window), geopolitical events have a positive impact on the Information Technology sector. Additionally, the Communication Services sector has an overall positive response to geopolitical risk in contrast to the Consumer Staples sector in the ($-3$,3), (0,1), and ($-10$,10) event windows. However, there is an opposite effect in the (0,5) event window where the Consumer Staples sector has a positive response to geopolitical risk in contrast to the Communication Services sector.

Another element of the implications of our study is that our results are ethically sound for the following reasons: First, we rigorously followed the seven systematic steps involved in event study during our data analysis exercise, such that our analysis is identical and consistent for all firms, leading to objective findings, conclusions and interpretations. Second, the use of publicly available data makes our study reproducible and avoids ethical implications that could arise as a result of the use of private data.

Our results also have some real-life and practical applications as follows: Companies within the Information Technology, Communication Services, and Consumer Staples sectors in the U.S. can intentionally study geopolitical events to determine how their stock prices could react whenever these uncertain events occur.

Furthermore, geopolitical risk is often a major cause of concern to company stakeholders, which includes policymakers, corporations, financial investors, politicians, etc. In particular, investors can decide to buy Information Technology stocks when a geopolitical event occurs due to the initial negative impact it has on the stock prices of the firms within

this sector (since the Information Technology firms' stocks will be trading at a discount compared to their future prices (for example, 10 days after the event occurs)).

*6.2. Shortcomings/Limitations and Suggestions for Future Research*

Although we do not see any obvious shortcomings from the methods used in our study, we do, however, observe some limitations, especially relating to the scope of our study. Some of these limitations are presented below and form the basis of future research.

We have employed the event study methodology and used the market model to measure the impact of geopolitical risk on the stock returns of the Information Technology and Communication Services, and Consumer Staples sectors' firms in the U.S. S&P 500 index. Future research could employ the Capital Asset Pricing Model (CAPM) (Campbell et al. 2012, pp. 181–84; Berk and DeMarzo 2017, pp. 417–24) instead to quantify the impact of geopolitical risk on the stock returns of the Information Technology, Communication Services, and Consumer Staples sectors. The results of the CAPM study can be compared with our results in an effort to determine whether similar or dissimilar results can be obtained.

Caldara and Iacoviello (2018) find that while the realization of adverse geopolitical events leads to smaller economic effects because the uncertainty is usually addressed and resolved, the shocks of geopolitical threats are usually protracted, leading to a rise in uncertainty and adverse economic activity. Future research might therefore include an event study measuring the impact of geopolitical risk on stocks, focusing on the events dates when the threats were initiated.

Modern technologies such as machine learning analysis, distributed ledger technology, block chains, and cryptocurrencies are currently being used to forecast the impact of certain risks on stock returns. Future research might include using these methods to investigate the impact of geopolitical risk on stock returns.

Finally, further studies may investigate the impact of geopolitical risk on the stocks of small and medium-sized enterprises (SMEs) in the U.S. stock market. Furthermore, additional research can be performed to evaluate the impact of geopolitical risk on the stock returns of other economic sectors within the S&P 500 index such as Health, Financials, Real Estate, etc. Additionally, this study is about the U.S. equity market, which has a western context. Further studies may research the impact of geopolitical risk on the stock returns of firms within a non-Western stock index such as the Shanghai Composite Index in China.

**Author Contributions:** G.A.F. and V.C.V. co-wrote the paper. A.M.M.S.Q. supervised the paper and contributed to the literature review, method, and data analysis. All authors have read and agreed to the published version of the manuscript.

**Funding:** This research received no external funding.

**Institutional Review Board Statement:** Not applicable.

**Informed Consent Statement:** Not applicable.

**Data Availability Statement:** Data is available on request from the authors.

**Conflicts of Interest:** The authors declare no conflict of interest.

# Appendix A

**Table A1.** Table of Descriptive Statistics for the normal returns of the 3 sectors ("Moscow theater hostage crisis" Geopolitical event) and event window ($-10,10$)).

| Tickers | Company | Sector | Mean | Std. Dev. | Maximum | Minimum |
|---|---|---|---|---|---|---|
| S&P 500 | S&P 500 Index | | 0.030% | 0.960% | 4.959% | −3.298% |
| ATVI | Activision Blizzard | Communication Services | 0.066% | 3.608% | 11.707% | −9.547% |
| GOOGL | Alphabet Inc. (Class A) | Communication Services | NA | NA | NA | NA |
| GOOG | Alphabet Inc. (Class C) | Communication Services | NA | NA | NA | NA |
| T | AT&T Inc. | Communication Services | −0.279% | 2.459% | 8.028% | −10.195% |
| CHTR | Charter Communications | Communication Services | NA | NA | NA | NA |
| CMCSA | Comcast Corp. | Communication Services | −0.234% | 3.413% | 15.257% | −12.890% |
| DISCA | Discovery, Inc. (Class A) | Communication Services | NA | NA | NA | NA |
| DISCK | Discovery, Inc. (Class C) | Communication Services | NA | NA | NA | NA |
| DISH | Dish Network | Communication Services | −0.078% | 3.228% | 13.217% | −10.884% |
| EA | Electronic Arts | Communication Services | 0.170% | 2.835% | 11.393% | −8.103% |
| FB | Facebook, Inc. | Communication Services | NA | NA | NA | NA |
| FOXA | Fox Corporation (Class A) | Communication Services | NA | NA | NA | NA |
| FOX | Fox Corporation (Class B) | Communication Services | NA | NA | NA | NA |
| IPG | Interpublic Group | Communication Services | −0.050% | 4.087% | 20.522% | −23.831% |
| LYV | Live Nation Entertainment | Communication Services | NA | NA | NA | NA |
| LUMN | Lumen Technologies | Communication Services | NA | NA | NA | NA |
| NFLX | Netflix Inc. | Communication Services | NA | NA | NA | NA |
| NWSA | News Corp. Class A | Communication Services | NA | NA | NA | NA |
| NWS | News Corp. Class B | Communication Services | NA | NA | NA | NA |
| OMC | Omnicom Group | Communication Services | −0.055% | 3.307% | 12.900% | −19.701% |
| TMUS | T-Mobile US | Communication Services | NA | NA | NA | NA |
| TTWO | Take-Two Interactive | Communication Services | 0.569% | 5.043% | 31.268% | −31.362% |
| TWTR | Twitter, Inc. | Communication Services | NA | NA | NA | NA |
| VZ | Verizon Communications | Communication Services | −0.174% | 2.504% | 9.272% | −11.846% |
| VIAC | ViacomCBS | Communication Services | NA | NA | NA | NA |

**Table A1.** *Cont.*

| Tickers | Company | Sector | Mean | Std. Dev. | Maximum | Minimum |
|---|---|---|---|---|---|---|
| DIS | The Walt Disney Company | Communication Services | −0.068% | 2.770% | 8.437% | −9.031% |
| MO | Altria Group Inc. | Consumer Staples | −0.11% | 2.01% | 5.44% | −11.40% |
| ADM | Archer-Daniels-Midland Co | Consumer Staples | 0.02% | 1.77% | 7.44% | −8.53% |
| BF.B | Brown-Forman Corp. | Consumer Staples | NA | NA | NA | NA |
| CPB | Campbell Soup | Consumer Staples | −0.07% | 1.58% | 4.99% | −6.10% |
| CHD | Church & Dwight | Consumer Staples | 0.11% | 1.96% | 9.31% | −7.25% |
| CLX | The Clorox Company | Consumer Staples | 0.06% | 1.69% | 5.87% | −5.35% |
| KO | Coca-Cola Company | Consumer Staples | 0.07% | 1.62% | 5.36% | −5.93% |
| CL | Colgate-Palmolive | Consumer Staples | −0.01% | 1.56% | 5.91% | −6.58% |
| CAG | Conagra Brands | Consumer Staples | 0.04% | 1.52% | 5.10% | −7.62% |
| STZ | Constellation Brands | Consumer Staples | 0.08% | 2.59% | 14.87% | −13.20% |
| COST | Costco Wholesale Corp. | Consumer Staples | −0.02% | 2.41% | 9.77% | −6.68% |
| EL | Estée Lauder Companies | Consumer Staples | −0.06% | 1.95% | 5.70% | −8.71% |
| GIS | General Mills | Consumer Staples | 0.02% | 1.67% | 4.17% | −10.59% |
| HSY | The Hershey Company | Consumer Staples | 0.01% | 2.13% | 25.28% | −11.94% |
| HRL | Hormel Foods Corp. | Consumer Staples | 0.00% | 1.68% | 5.25% | −7.30% |
| SJM | JM Smucker | Consumer Staples | 0.11% | 2.49% | 20.32% | −4.82% |
| K | Kellogg Co. | Consumer Staples | 0.09% | 1.63% | 7.43% | −3.75% |
| KMB | Kimberly-Clark | Consumer Staples | −0.03% | 1.45% | 5.68% | −7.46% |
| KHC | Kraft Heinz Co | Consumer Staples | NA | NA | NA | NA |
| KR | Kroger Co. | Consumer Staples | −0.23% | 2.09% | 5.65% | −14.51% |
| LW | Lamb Weston Holdings Inc. | Consumer Staples | NA | NA | NA | NA |
| MKC | McCormick & Co. | Consumer Staples | 0.04% | 1.59% | 6.99% | −5.11% |
| TAP | Molson Coors Beverage Company | Consumer Staples | 0.13% | 1.71% | 7.96% | −7.28% |
| MDLZ | Mondelez International | Consumer Staples | 0.05% | 1.68% | 5.98% | −7.41% |
| MNST | Monster Beverage | Consumer Staples | 0.10% | 3.10% | 13.33% | −16.00% |
| PEP | PepsiCo Inc. | Consumer Staples | −0.05% | 2.02% | 14.87% | −10.17% |
| PM | Philip Morris International | Consumer Staples | NA | NA | NA | NA |
| PG | Procter & Gamble | Consumer Staples | 0.09% | 1.51% | 4.53% | −7.38% |
| SYY | Sysco Corp. | Consumer Staples | 0.07% | 1.81% | 6.83% | −5.43% |
| TSN | Tyson Foods | Consumer Staples | 0.07% | 2.78% | 12.97% | −9.16% |
| WMT | Walmart | Consumer Staples | 0.02% | 1.94% | 8.03% | −6.66% |
| WBA | Walgreens Boots Alliance | Consumer Staples | −0.01% | 1.90% | 6.01% | −9.24% |
| ACN | Accenture plc | Information Technology | 0.000554 | 0.038213 | 0.108998 | −0.11842 |
| ADBE | Adobe Inc. | Information Technology | −0.00053 | 0.042037 | 0.139202 | −0.29758 |
| AMD | Advanced Micro Devices Inc. | Information Technology | −0.00208 | 0.052198 | 0.181818 | −0.32402 |

**Table A1.** *Cont.*

| Tickers | Company | Sector | Mean | Std. Dev. | Maximum | Minimum |
|---|---|---|---|---|---|---|
| AKAM | Akamai Technologies Inc. | Information Technology | −0.00453 | 0.06334 | 0.22884 | −0.19205 |
| APH | Amphenol Corp | Information Technology | −0.00064 | 0.024493 | 0.101449 | −0.08636 |
| ADI | Analog Devices, Inc. | Information Technology | −0.00203 | 0.040635 | 0.11771 | −0.0996 |
| ANSS | ANSYS | Information Technology | 0.000129 | 0.039131 | 0.132035 | −0.10498 |
| AAPL | Apple Inc. | Information Technology | −0.00014 | 0.031971 | 0.084557 | −0.15037 |
| AMAT | Applied Materials Inc. | Information Technology | −0.00077 | 0.042385 | 0.145161 | −0.14018 |
| ANET | Arista Networks | Information Technology | NA | NA | NA | NA |
| ADSK | Autodesk Inc. | Information Technology | −0.00089 | 0.031786 | 0.095938 | −0.18718 |
| ADP | Automatic Data Processing | Information Technology | −0.00094 | 0.024506 | 0.057002 | −0.23579 |
| AVGO | Broadcom Inc. | Information Technology | NA | NA | NA | NA |
| BR | Broadridge Financial Solutions | Information Technology | NA | NA | NA | NA |
| CDNS | Cadence Design Systems | Information Technology | −0.00187 | 0.033792 | 0.121036 | −0.18216 |
| CDW | CDW | Information Technology | NA | NA | NA | NA |
| CSCO | Cisco Systems | Information Technology | −0.00141 | 0.039594 | 0.243884 | −0.11306 |
| CTXS | Citrix Systems | Information Technology | −0.00444 | 0.043899 | 0.156762 | −0.17753 |
| CTSH | Cognizant Technology Solutions | Information Technology | 0.003875 | 0.039919 | 0.186666 | −0.16814 |
| GLW | Corning Inc. | Information Technology | −0.00589 | 0.057967 | 0.183024 | −0.35223 |
| DXC | DXC Technology | Information Technology | −0.00037 | 0.033476 | 0.180907 | −0.16098 |
| FFIV | F5 Networks | Information Technology | $6.13 \times 10^{-5}$ | 0.060255 | 0.155556 | −0.2027 |
| FIS | Fidelity National Information Services | Information Technology | −0.00073 | 0.031224 | 0.123341 | −0.32746 |
| FISV | Fiserv Inc. | Information Technology | −0.00104 | 0.027776 | 0.089539 | −0.13368 |
| FLT | FleetCor Technologies Inc. | Information Technology | NA | NA | NA | NA |
| FLIR | FLIR Systems | Information Technology | −0.00023 | 0.03921 | 0.137449 | −0.16223 |
| FTNT | Fortinet | Information Technology | NA | NA | NA | NA |
| IT | Gartner Inc. | Information Technology | −0.00074 | 0.035787 | 0.13396 | −0.11917 |
| GPN | Global Payments Inc. | Information Technology | −0.00027 | 0.030708 | 0.111255 | −0.14912 |
| HPE | Hewlett Packard Enterprise | Information Technology | NA | NA | NA | NA |

**Table A1.** *Cont.*

| Tickers | Company | Sector | Mean | Std. Dev. | Maximum | Minimum |
|---|---|---|---|---|---|---|
| HPQ | HP Inc. | Information Technology | −0.0009 | 0.031923 | 0.172883 | −0.06552 |
| INTC | Intel Corp. | Information Technology | −0.00132 | 0.037014 | 0.108222 | −0.18519 |
| IBM | International Business Machines | Information Technology | −0.00187 | 0.023209 | 0.077778 | −0.10118 |
| INTU | Intuit Inc. | Information Technology | 0.001386 | 0.029649 | 0.108777 | −0.07632 |
| IPGP | IPG Photonics Corp. | Information Technology | NA | NA | NA | NA |
| JKHY | Jack Henry & Associates | Information Technology | −0.00299 | 0.034398 | 0.128923 | −0.26792 |
| JNPR | Juniper Networks | Information Technology | −0.00311 | 0.058029 | 0.265625 | −0.18442 |
| KEYS | Keysight Technologies | Information Technology | NA | NA | NA | NA |
| KLAC | KLA Corporation | Information Technology | $-5.6 \times 10^{-5}$ | 0.040825 | 0.154486 | −0.09582 |
| LRCX | Lam Research | Information Technology | −0.00262 | 0.044031 | 0.122459 | −0.14244 |
| LDOS | Leidos Holdings | Information Technology | NA | NA | NA | NA |
| MA | Mastercard Inc. | Information Technology | NA | NA | NA | NA |
| MXIM | Maxim Integrated Products Inc. | Information Technology | −0.00168 | 0.041277 | 0.121053 | −0.11819 |
| MCHP | Microchip Technology | Information Technology | 0.000976 | 0.040282 | 0.140351 | −0.10277 |
| MU | Micron Technology | Information Technology | −0.00102 | 0.0464 | 0.128771 | −0.14915 |
| MSFT | Microsoft Corp. | Information Technology | −0.00063 | 0.02654 | 0.111178 | −0.09472 |
| MSI | Motorola Solutions Inc. | Information Technology | −0.00163 | 0.032905 | 0.109174 | −0.10509 |
| NTAP | NetApp | Information Technology | −0.00014 | 0.064495 | 0.276596 | −0.16574 |
| NLOK | NortonLifeLock | Information Technology | NA | NA | NA | NA |
| NVDA | Nvidia Corporation | Information Technology | −0.00428 | 0.054642 | 0.183349 | −0.31751 |
| ORCL | Oracle Corp. | Information Technology | −0.00144 | 0.039702 | 0.113768 | −0.14509 |
| PAYX | Paychex Inc. | Information Technology | −0.00073 | 0.029368 | 0.097397 | −0.13182 |
| PAYC | Paycom | Information Technology | NA | NA | NA | NA |
| PYPL | PayPal | Information Technology | NA | NA | NA | NA |
| QRVO | Qorvo | Information Technology | NA | NA | NA | NA |
| QCOM | QUALCOMM Inc. | Information Technology | −0.00032 | 0.041167 | 0.186694 | −0.11856 |
| CRM | Salesforce.com | Information Technology | NA | NA | NA | NA |
| STX | Seagate Technology | Information Technology | NA | NA | NA | NA |

**Table A1.** *Cont.*

| Tickers | Company | Sector | Mean | Std. Dev. | Maximum | Minimum |
|---|---|---|---|---|---|---|
| NOW | ServiceNow | Information Technology | NA | NA | NA | NA |
| SWKS | Skyworks Solutions | Information Technology | −0.00384 | 0.069476 | 0.242924 | −0.20875 |
| SNPS | Synopsys Inc. | Information Technology | −0.00043 | 0.028026 | 0.104531 | −0.08713 |
| TEL | TE Connectivity Ltd. | Information Technology | NA | NA | NA | NA |
| TER | Teradyne | Information Technology | −0.00305 | 0.045721 | 0.140481 | −0.15034 |
| TXN | Texas Instruments | Information Technology | −0.00192 | 0.037069 | 0.112323 | −0.08863 |
| TYL | Tyler Technologies | Information Technology | 0.001719 | 0.039332 | 0.14375 | −0.11837 |
| VRSN | Verisign Inc. | Information Technology | −0.00706 | 0.068025 | 0.27619 | −0.45779 |
| V | Visa Inc. | Information Technology | NA | NA | NA | NA |
| VNT | Vontier | Information Technology | NA | NA | NA | NA |
| WDC | Western Digital | Information Technology | 0.003972 | 0.057895 | 0.277473 | −0.20918 |
| WU | Western Union Co | Information Technology | NA | NA | NA | NA |
| XRX | Xerox | Information Technology | −0.00098 | 0.043395 | 0.158878 | −0.25 |
| XLNX | Xilinx | Information Technology | −0.00141 | 0.048801 | 0.150455 | −0.12053 |
| ZBRA | Zebra Technologies | Information Technology | 0.001042 | 0.024991 | 0.163879 | −0.06394 |

**Table A2.** List of geopolitical events 1962–2020.

| Name | Start Date (MM/DD/YYYY) |
|---|---|
| Cuban Missile Crisis | 10/1/1962 |
| Escalation of Vietnam War | 2/1/1963 |
| Tensions Middle East pre Six-Day War | 5/1/1967 |
| Six Day War | 6/1/1967 |
| Tet Offensive | 2/1/1968 |
| Vietnam War: Operation Jefferson Glenn | 9/1/1970 |
| Vietnam War: First Battle of Quảng Trị | 5/1/1972 |
| Munich Massacre | 9/1/1972 |
| Yom Kippur War | 10/1/1973 |
| Grain Embargo against USSR | 1/1/1980 |
| Falklands War begins | 4/1/1982 |
| Lebanon and Falklands War | 6/1/1982 |
| Able Archer 83 | 12/1/1983 |
| TWA Hijacking | 6/1/1985 |
| Rome and Vienna terror attacks | 1/1/1986 |
| US bombs Lybia | 4/15/1986 |
| Tear down this wall speech | 6/12/1987 |
| Mecca incident | 8/1/1987 |
| Lockerbie bombing | 12/21/1988 |
| Fall of Berlin Wall | 11/9/1989 |
| Revolutions of 1989 | 12/1/1989 |
| Kuwait Invasion | 8/2/1990 |
| Gulf War | 8/2/1990 |

**Table A2.** *Cont.*

| Name | Start Date (MM/DD/YYYY) |
|---|---|
| Gorbachev ousted | 10/1/1991 |
| Oklahoma City Bombing | 4/19/1995 |
| Taiwan strait crisis | 2/1/1996 |
| Iraq refuses UN inspectors | 3/1/1996 |
| Iraq disarmament crisis | 7/1/1996 |
| 1998 U.S. embassy bombings | 8/7/1998 |
| Russian apartment bombings | 9/4/1999 |
| USS Cole Bombings, 2nd Intifada | 10/12/2000 |
| "9/11" | 9/11/2001 |
| U.S. Invasion of Afghanistan | 10/7/2001 |
| U.S. names sponsors of terror | 3/1/2002 |
| Moscow theater hostage crisis | 10/23/2002 |
| Moscow theater hostage crisis | 11/1/2002 |
| Before Iraq Invasion | 2/1/2003 |
| Iraq Invasion | 3/20/2003 |
| Madrid bombings | 3/11/2004 |
| Beslan school siege | 9/1/2004 |
| London bombings | 7/7/2005 |
| Tensions over Iran and nuclear treaty | 7/1/2006 |
| 2006 transatlantic aircraft plot | 8/1/2006 |
| Obama announces surge Afghanistan | 12/1/2009 |
| Arab Spring: Mubarak resigns | 2/11/2011 |
| Arab Spring: Syrian Civil War and Lybian Civil War | 3/1/2011 |
| North Korean satellite explodes | 4/12/2012 |
| Boston Marathon Bombings | 4/15/2013 |
| Westgate shopping mall attack, Nairobi | 9/21/2013 |
| Russia annexes Crimea | 3/1/2014 |
| Missile hits plane in Ukraine | 7/1/2014 |
| Ukraine and ISIS | 8/1/2014 |
| Paris Attacks | 11/13/2015 |
| San Bernardino Shootings | 12/2/2015 |
| Syrian tensions | 4/1/2018 |
| U.S.–Iran tensions | 7/1/2018 |
| U.S.–Iran tensions | 6/1/2019 |
| U.S.–China tensions | 8/1/2019 |

**Table A3.** Average abnormal returns and cumulative average abnormal returns for the Information Technology, Communication Services, and Consumer Staples sectors for the "Moscow theater hostage crisis" geopolitical event and event window (−10,10).

| | "Moscow Theater Hostage Crisis" Geopolitical Event | | | | | |
|---|---|---|---|---|---|---|
| | Information Technology | | Communication Services | | Consumer Staples | |
| Event Day | AAR | CAAR | AAR | CAAR | AAR | CAAR |
| −10 | 2.614% | 2.614% | 1.293% | 1.293% | −0.972% | −0.972% |
| −9 | 1.004% | 3.618% | 1.281% | 2.573% | −0.756% | −1.728% |
| −8 | 1.133% | 4.751% | −0.609% | 1.964% | −0.147% | −1.875% |
| −7 | 0.039% | 4.790% | −0.854% | 1.110% | 1.346% | −0.529% |
| −6 | 0.112% | 4.902% | −2.391% | −1.281% | −0.826% | −1.355% |
| −5 | −2.509% | 2.394% | 0.690% | −0.591% | −0.253% | −1.608% |
| −4 | 3.111% | 5.505% | −4.123% | −4.714% | −0.774% | −2.383% |
| −3 | 0.273% | 5.778% | −0.598% | −5.312% | 0.606% | −1.777% |
| −2 | 3.013% | 8.792% | 0.645% | −4.667% | 1.063% | −0.714% |
| −1 | −1.099% | 7.692% | 0.857% | −3.810% | −1.153% | −1.867% |
| 0 | 2.410% | 10.102% | 1.769% | −2.041% | −0.148% | −2.015% |
| 1 | 1.195% | 11.297% | 0.080% | −1.961% | −0.408% | −2.422% |
| 2 | 0.611% | 11.908% | −2.031% | −3.992% | −0.080% | −2.502% |

**Table A3.** *Cont.*

| | "Moscow Theater Hostage Crisis" Geopolitical Event | | | | | |
| | Information Technology | | Communication Services | | Consumer Staples | |
| Event Day | AAR | CAAR | AAR | CAAR | AAR | CAAR |
|---|---|---|---|---|---|---|
| 3 | 0.647% | 12.555% | −1.360% | −5.352% | −1.391% | −3.893% |
| 4 | −1.706% | 10.848% | 0.496% | −4.856% | 1.302% | −2.591% |
| 5 | 2.045% | 12.893% | 0.858% | −3.998% | −0.656% | −3.248% |
| 6 | 1.182% | 14.075% | 0.075% | −3.923% | 0.185% | −3.063% |
| 7 | 2.159% | 16.234% | 0.614% | −3.309% | −0.318% | −3.381% |
| 8 | 2.642% | 18.876% | 2.759% | −0.550% | −2.418% | −5.799% |
| 9 | −1.258% | 17.618% | −0.105% | −0.655% | 0.099% | −5.699% |
| 10 | 1.959% | 19.577% | 0.053% | −0.603% | −0.698% | −6.398% |

**Table A4.** Number of events per firm in the Information Technology, Communication Services, and Consumer Staples sectors included in the event study.

| Tickers | Company | Sector | Number of Events |
|---|---|---|---|
| ATVI | Activision Blizzard | Communication Services | 34 |
| GOOGL | Alphabet Inc. (Class A) | Communication Services | 19 |
| GOOG | Alphabet Inc. (Class C) | Communication Services | 19 |
| T | AT&T Inc. | Communication Services | 46 |
| CHTR | Charter Communications | Communication Services | 14 |
| CMCSA | Comcast Corp. | Communication Services | 48 |
| DISCA | Discovery, Inc. (Class A) | Communication Services | 17 |
| DISCK | Discovery, Inc. (Class C) | Communication Services | 15 |
| DISH | Dish Network | Communication Services | 33 |
| EA | Electronic Arts | Communication Services | 39 |
| FB | Facebook, Inc. | Communication Services | 11 |
| FOXA | Fox Corporation (Class A) | Communication Services | 2 |
| FOX | Fox Corporation (Class B) | Communication Services | 2 |
| IPG | Interpublic Group | Communication Services | 48 |
| LYV | Live Nation Entertainment | Communication Services | 17 |
| LUMN | Lumen Technologies | Communication Services | 0 |
| NFLX | Netflix Inc. | Communication Services | 24 |
| NWSA | News Corp. Class A | Communication Services | 10 |
| NWS | News Corp. Class B | Communication Services | 10 |
| OMC | Omnicom Group | Communication Services | 48 |
| TMUS | T-Mobile US | Communication Services | 15 |
| TTWO | Take-Two Interactive | Communication Services | 30 |
| TWTR | Twitter, Inc. | Communication Services | 9 |
| VZ | Verizon Communications | Communication Services | 46 |
| VIAC | ViacomCBS | Communication Services | 17 |
| DIS | The Walt Disney Company | Communication Services | 58 |
| MO | Altria Group Inc. | Consumer Staples | 58 |
| ADM | Archer-Daniels-Midland Co. | Consumer Staples | 48 |
| BF.B | Brown-Forman Corp. | Consumer Staples | 48 |
| CPB | Campbell Soup | Consumer Staples | 50 |
| CHD | Church & Dwight | Consumer Staples | 48 |
| CLX | The Clorox Company | Consumer Staples | 50 |
| KO | Coca-Cola Company | Consumer Staples | 58 |
| CL | Colgate-Palmolive | Consumer Staples | 50 |
| CAG | Conagra Brands | Consumer Staples | 48 |
| STZ | Constellation Brands | Consumer Staples | 34 |
| COST | Costco Wholesale Corp. | Consumer Staples | 42 |
| EL | Estée Lauder Companies | Consumer Staples | 33 |
| GIS | General Mills | Consumer Staples | 48 |
| HSY | The Hershey Company | Consumer Staples | 48 |

**Table A4.** *Cont.*

| Tickers | Company | Sector | Number of Events |
|---|---|---|---|
| HRL | Hormel Foods Corp. | Consumer Staples | 48 |
| SJM | JM Smucker | Consumer Staples | 34 |
| K | Kellogg Co. | Consumer Staples | 50 |
| KMB | Kimberly Clark | Consumer Staples | 48 |
| KHC | Kraft Heinz Co. | Consumer Staples | 6 |
| KR | Kroger Co. | Consumer Staples | 49 |
| LW | Lamb Weston Holdings Inc. | Consumer Staples | 4 |
| MKC | McCormick & Co. | Consumer Staples | 50 |
| TAP | Molson Coors Beverage Company | Consumer Staples | 49 |
| MDLZ | Mondelez International | Consumer Staples | 27 |
| MNST | Monster Beverage | Consumer Staples | 44 |
| PEP | PepsiCo Inc. | Consumer Staples | 51 |
| PM | Philip Morris International | Consumer Staples | 15 |
| PG | Procter & Gamble | Consumer Staples | 58 |
| SYY | Sysco Corp. | Consumer Staples | 50 |
| TSN | Tyson Foods | Consumer Staples | 48 |
| WMT | Walmart | Consumer Staples | 51 |
| WBA | Walgreens Boots Alliance | Consumer Staples | 48 |
| CAN | Accenture plc | Information Technology | 27 |
| ADBE | Adobe Inc. | Information Technology | 42 |
| AMD | Advanced Micro Devices Inc. | Information Technology | 48 |
| AKAM | Akamai Technologies Inc. | Information Technology | 28 |
| APH | Amphenol Corp. | Information Technology | 34 |
| ADI | Analog Devices, Inc. | Information Technology | 48 |
| ANSS | ANSYS | Information Technology | 31 |
| AAPL | Apple Inc. | Information Technology | 48 |
| AMAT | Applied Materials Inc. | Information Technology | 48 |
| ANET | Arista Networks | Information Technology | 8 |
| ADSK | Autodesk Inc. | Information Technology | 44 |
| ADP | Automatic Data Processing | Information Technology | 48 |
| AVGO | Broadcom Inc. | Information Technology | 15 |
| BR | Broadridge Financial Solutions | Information Technology | 15 |
| CDNS | Cadence Design Systems | Information Technology | 42 |
| CDW | CDW | Information Technology | 10 |
| CSCO | Cisco Systems | Information Technology | 37 |
| CTXS | Citrix Systems | Information Technology | 33 |
| CTSH | Cognizant Technology Solutions | Information Technology | 30 |
| GLW | Corning Inc. | Information Technology | 48 |
| DXC | DXC Technology | Information Technology | 51 |
| FFIV | F5 Networks | Information Technology | 29 |
| FIS | Fidelity National Information Services | Information Technology | 27 |
| FISV | Fiserv Inc. | Information Technology | 42 |
| FLT | FleetCor Technologies Inc. | Information Technology | 14 |
| FLIR | FLIR Systems | Information Technology | 34 |
| FTNT | Fortinet | Information Technology | 15 |
| IT | Gartner Inc. | Information Technology | 34 |
| GPN | Global Payments Inc. | Information Technology | 27 |
| HPE | Hewlett Packard Enterprise | Information Technology | 6 |
| HPQ | HP Inc. | Information Technology | 58 |
| INTC | Intel Corp. | Information Technology | 48 |
| IBM | International Business Machines | Information Technology | 58 |
| INTU | Intuit Inc. | Information Technology | 34 |
| IPGP | IPG Photonics Corp. | Information Technology | 15 |
| JKHY | Jack Henry & Associates | Information Technology | 44 |
| JNPR | Juniper Networks | Information Technology | 29 |
| KEYS | Keysight Technologies | Information Technology | 6 |

**Table A4.** *Cont.*

| Tickers | Company | Sector | Number of Events |
|---|---|---|---|
| KLAC | KLA Corporation | Information Technology | 48 |
| LRCX | Lam Research | Information Technology | 45 |
| LDOS | Leidos Holdings | Information Technology | 15 |
| MA | Mastercard Inc. | Information Technology | 17 |
| MXIM | Maxim Integrated Products Inc. | Information Technology | 40 |
| MCHP | Microchip Technology | Information Technology | 34 |
| MU | Micron Technology | Information Technology | 45 |
| MSFT | Microsoft Corp. | Information Technology | 43 |
| MSI | Motorola Solutions Inc. | Information Technology | 49 |
| NTAP | NetApp | Information Technology | 33 |
| NLOK | NortonLifeLock | Information Technology | 2 |
| NVDA | Nvidia Corporation | Information Technology | 29 |
| ORCL | Oracle Corp. | Information Technology | 43 |
| PAYX | Paychex Inc. | Information Technology | 46 |
| PAYC | Paycom | Information Technology | 8 |
| PYPL | PayPal | Information Technology | 6 |
| QRVO | Qorvo | Information Technology | 6 |
| QCOM | QUALCOMM Inc. | Information Technology | 34 |
| CRM | Salesforce.com | Information Technology | 19 |
| STX | Seagate Technology | Information Technology | 22 |
| NOW | ServiceNow | Information Technology | 11 |
| SWKS | Skyworks Solutions | Information Technology | 45 |
| SNPS | Synopsys Inc. | Information Technology | 34 |
| TEL | TE Connectivity Ltd. | Information Technology | 15 |
| TER | Teradyne | Information Technology | 50 |
| TXN | Texas Instruments | Information Technology | 51 |
| TYL | Tyler Technologies | Information Technology | 48 |
| VRSN | Verisign Inc. | Information Technology | 30 |
| V | Visa Inc. | Information Technology | 15 |
| VNT | Vontier | Information Technology | 0 |
| WDC | Western Digital | Information Technology | 49 |
| WU | Western Union Co. | Information Technology | 15 |
| XRX | Xerox | Information Technology | 49 |
| XLNX | Xilinx | Information Technology | 37 |
| ZBRA | Zebra Technologies | Information Technology | 35 |

**Table A5.** Summary of event study regressions needed vs. possible regressions for the S&P 500 index.

| Number of Event Windows | 4 | | | |
|---|---|---|---|---|
| Number of Geopolitical Events | 58 | | | |
| Sectors in the S&P 500 Index | Number of Industries Per Sector | Number of Firms Per Sector | Possible Regressions Needed | Actual Number of Regressions Needed |
| Communication Services | 10 | 26 | 6032 | 2524 |
| Consumer Discretionary | 23 | 61 | 14,152 | 8108 |
| Consumer Staples | 12 | 32 | 7424 | 5572 |
| Energy | 5 | 25 | 5800 | 3864 |
| Financials | 12 | 65 | 15,080 | 9828 |
| Health Care | 10 | 63 | 14,616 | 9424 |
| Industrials | 17 | 73 | 16,936 | 11,104 |
| Information Technology | 13 | 73 | 16,936 | 9292 |
| Materials | 11 | 28 | 6496 | 4004 |
| Real Estate | 8 | 31 | 7192 | 4356 |
| Utilities | 5 | 28 | 6496 | 5064 |
| Total | 126 | 505 | 117,160 | 73,140 |

**Table A6.** Summary of event study regressions needed vs. possible regressions for the Information Technology, Communication Services, and Consumer Staples sectors of the S&P 500 index.

| Number of Event Windows | 4 | | | | |
|---|---|---|---|---|---|
| Number of Geopolitical Events | 58 | | | | |

| Sectors in the S&P 500 Index | Sector Randomly Selected | Number of Industries Analyzed | Number of Firms Analyzed | Possible Regressions Needed | Actual Number of Regressions Performed |
|---|---|---|---|---|---|
| Communication Services | Yes | 10 | 26 | 6032 | 2524 |
| Consumer Discretionary | No | | | | |
| Consumer Staples | Yes | 12 | 32 | 7424 | 5572 |
| Energy | No | | | | |
| Financials | No | | | | |
| Health Care | No | | | | |
| Industrials | No | | | | |
| Information Technology | Yes | 13 | 73 | 16,936 | 9292 |
| Materials | No | | | | |
| Real Estate | No | | | | |
| Utilities | No | | | | |
| Total | 3 | 35 | 131 | 30,392 | 17,388 |

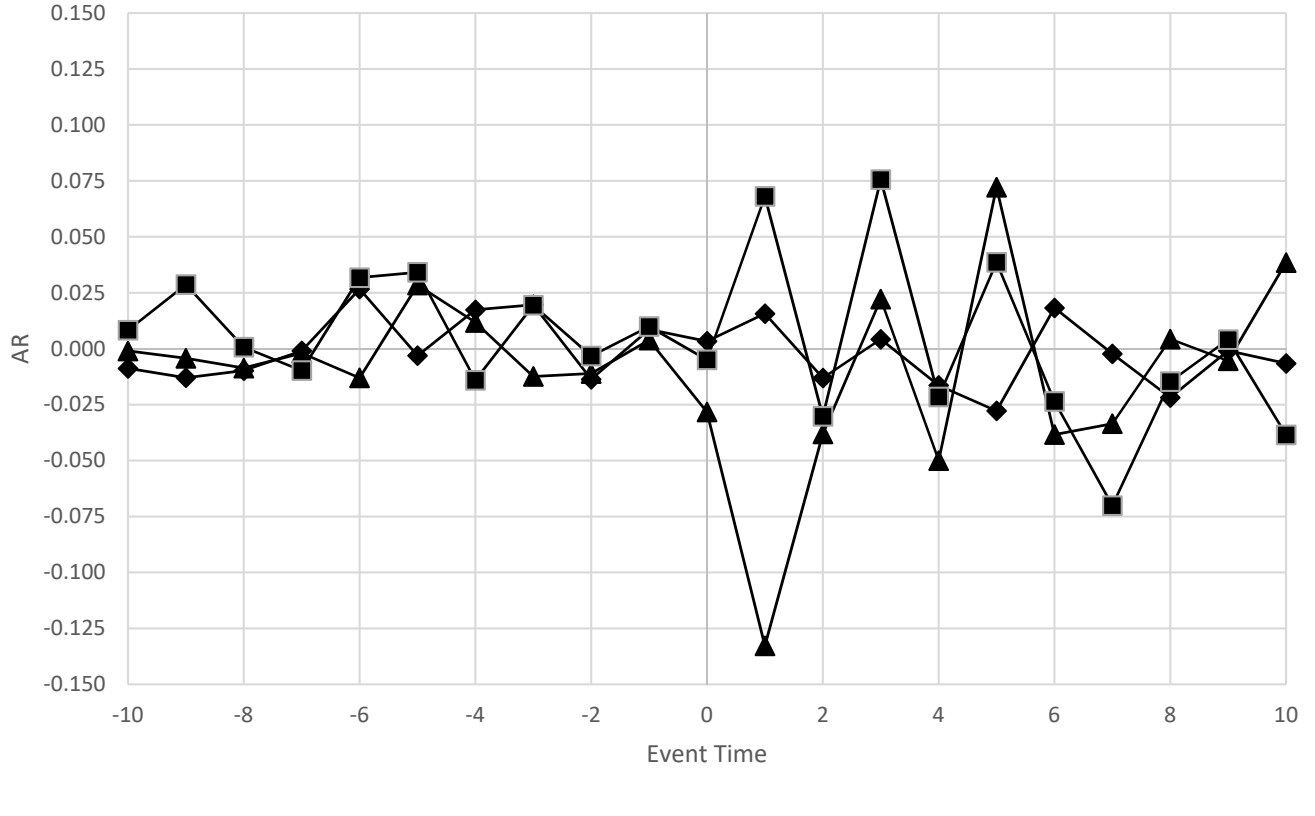

**Figure A1.** Plot of abnormal returns for the "9/11" Geopolitical event and event window (−10,10).

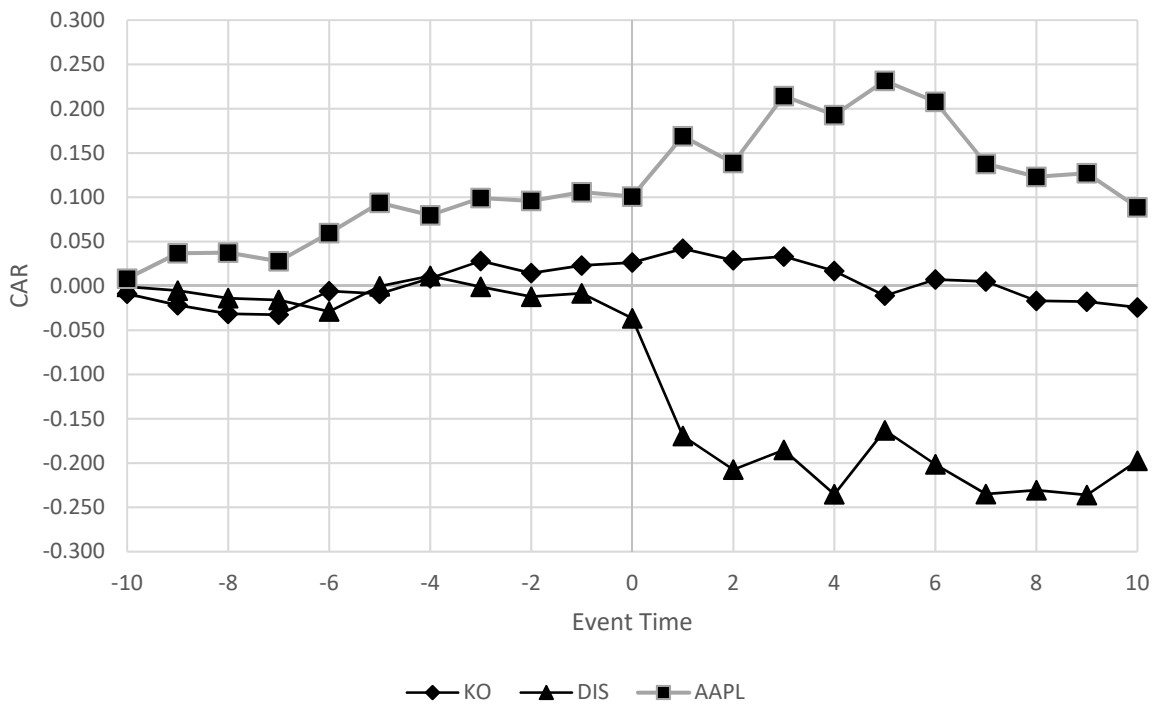

**Figure A2.** Plot of cumulative abnormal returns for the "9/11" Geopolitical event and event window (−10,10).

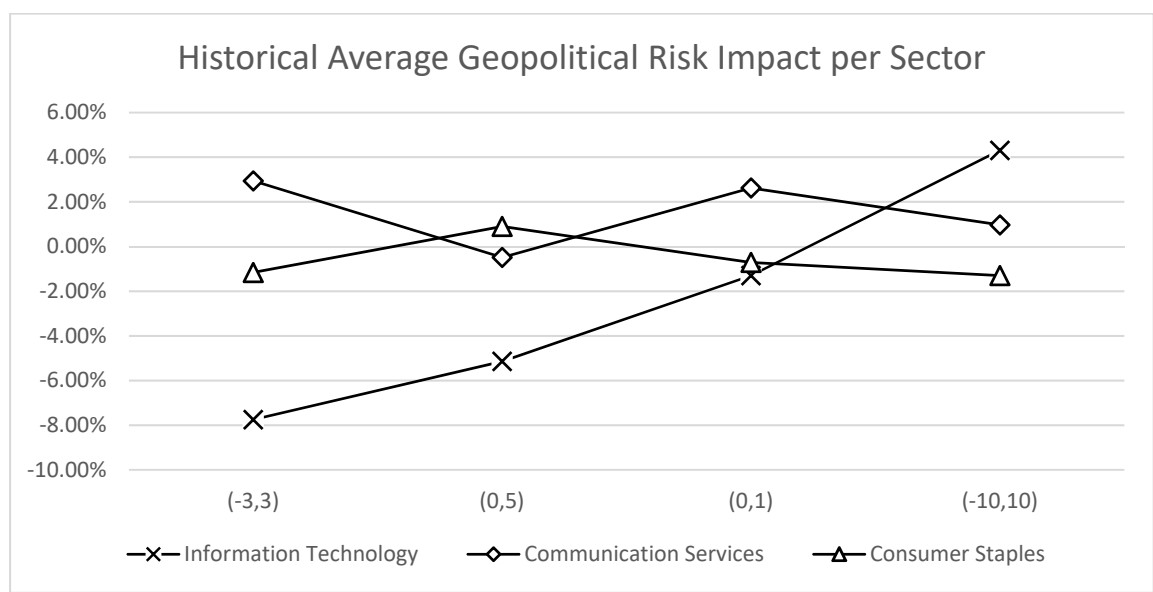

**Figure A3.** Overview of the impact of geopolitical risk (Historical Sector Average CAAR) on the Information Technology, Communication Services, and Consumer Staples sectors.

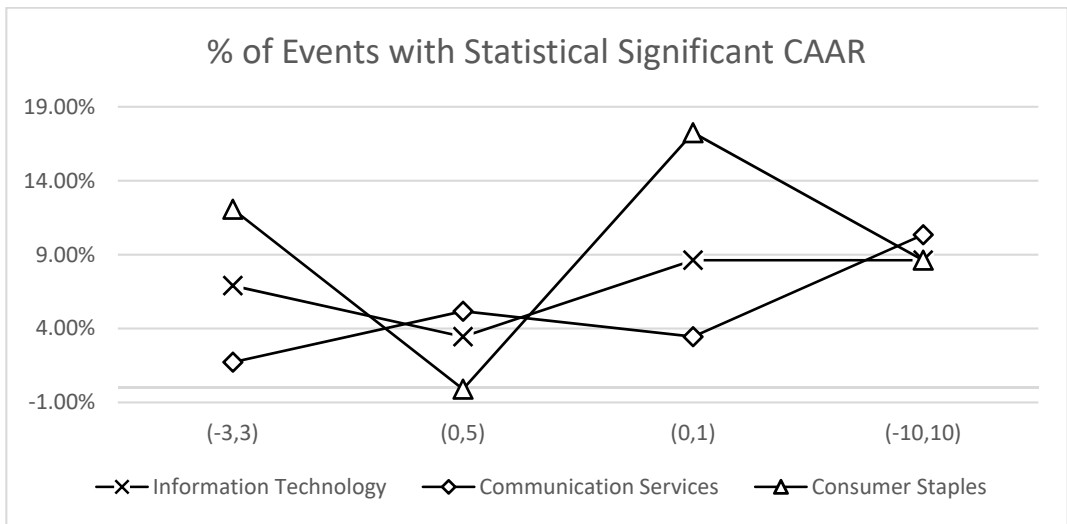

**Figure A4.** Summary of events with statistically significant CAARs for the Information Technology, Communication Services, and Consumer Staples sectors.

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
