# Peer review of "Impact of Geopolitical Risk on the Information Technology, Communication Services and Consumer Staples Sectors of the S&P 500 Index"

_jrfm, doi:10.3390/jrfm14110552_

Round 1
Reviewer 1 Report
Thank you for your interesting study.
I think that your work needs to be revised considerably.
First of all, I urge you to check the author information, because it seems to me that you have not complied with all the layout rules and in addition you have included your own names!
Second of all, you have submitted in a spacial issue which states "Studies focusing on the contributions to the analysis of machine learning, high frequency trading, distributed ledger technology, block chain, cryptocurrency, peer to peer, and so on are especially encouraged. The Financial Technology and Innovation section is open to publishing new and challenging studies focusing on a single country or a group of countries" I do not notice that you have properly developed these topics.
Finally, I recommend that you specify the practical implications, what is the practical problem that your paper wants to solve? To date, I do not understand it.
Author Response
This is an empirical study where we study the effects of geopolitical risk on the returns of the Information Technology, Communication Services, and Consumer Staples sector firms within the S&P 500 index.
The study is conducted rigorously. Moreover, we have now revised the literature section and made it shorter.
We do not mind publishing under any suitable topic of this journal.
Reviewer 2 Report
The authors undertook a huge job related to identifying the impact of geopolitical events on the companies' shares in three sectors. The job required huge data collection due to the event window identification. There are several concerns I suggest consider:
- The article is very long, mainly due to the tables presented in the appendix. In many cases, data are not available in the tables. I strongly recommend to re-arrange the tables.
- In both sections, the introduction and literature review authors present some papers from the literature. It is a bit chaotic for the reader. Furthermore, the presentation of the topic coming from the literature is not systematic. First, the authors report various empirical findings and then they mention Fama's work of 1970 which constitutes efficient market hypothesis being a basis for many analyses in finance. Fama's work refers to the authors' findings as well. If there is an impact of events on returns, can we still talk about semi-strong market efficiency? Therefore, the literature review must be systematized into the theoretical background of the analysis and most relevant empirical findings.
- The authors employed a standard event study procedure. Although testing procedure (formulas 13 and 14) is based on the assumption of normality. The t statistic is based on assuming Gaussian distribution. No testing for normality results is presented in the paper. It is obvious because financial returns based on daily data are in general not normal. The distributions are leptokurtic and their tails are fatter than normal. Thus the question of results robustness arises. There are bootstrap-based procedures that allow a robustness check of events study results. Otherwise, the paper will not contribute more than the others. There fore in the discussion we can read: Our results are in tandem with... but they do contradict with... The authors made a huge effort to examine so many events, therefore they should care about their work to be powerful for the methodology applied.
Author Response
1. Main reasoning around keeping the “N/A” values on the Appendix (Tables A1, A5, A6, A7) is to visualize the cases where the abnormal returns have not been statistical significant. In particular, the fact that there are so many cases with no statistical results is also visualized in Figure A4. It is made clear that only 10-15% of results are statistically significant depending on the sector and window. Given the large number of regressions taking place in this study, it’s a way to giving to the reader a better understanding to the output results.
- We have revised the literature sections. The focus of the study is not finding whether the market is efficient or not. We wanted to capture the effect of geopolitical risk. Hence we designed the literature review section in that context.
- As visualized in the Appendix (Figure A4), 20% of the actual CAAR results have been classified as statistically significant. That means that if a result does not pass the t-Test with more or equal to 1.96 then it is not included in our results. As mentioned in the end of Chapter 5, we have enhanced the robustness of the results by using the same parameters and procedures to a large number of firms from different sectors as well testing over a number of different event windows resulting in a very large number of regressions (more than 17,000). Therefore, even if there were a number of results that failed the t-Test marginally, the law of large numbers and the central limit theorem secures the robustness of our results.
Furthermore, we already discuss the robustness of our results in section 4 of this paper as follows: “Second, we limited potential bias and improved the reliability & robustness of our results by performing the event study that applies the same parameters and repeats the same procedures on firms in the Information Technology, Communication Services, and Consumer Staples sectors.”, and in section 6 of our paper as follows: “We also employed the quantitative empirical analysis as the preferred method in our study to analyze the data and followed the 7 systematic steps of the event study methodology to quantitatively measure the operationalized variables (across multiple event windows with different interval durations). The above approaches strengthened the validity, robustness, reliability, ethics as well as the soundness of our results.”
Reviewer 3 Report
Dear Authors,
Please find below and attached my comments and suggestions for your work.
Good luck!
Kind regards,
The Reviewer
Review Report Form
Journal: JRFM (ISSN 1911-8074)
Manuscript ID: jrfm-1408788
Type: Article
Title: Impact of Geopolitical Risk on the Information Technology, Communication Services & Consumer Staples sectors of the S&P 500 index
Authors: Gerard Atabong Fossung , Vasileios Chatzis Vovas , A.M.M. Shahiduzzaman Quoreshi *
Submission Date: 20 September 2021
Dear Authors,
I have carefully analyzed your article entitled “Impact of Geopolitical Risk on the Information Technology, Communication Services & Consumer Staples sectors of the S&P 500 index”.
Congratulations for your work and valuable insights reflected in the content of the manuscript!
The structure of my Review Report Form takes into consideration two sections, namely: (A.) General overview of the article and strong points; and (B) Suggestions meant to improve your current manuscript.
(A.) General overview of the article and strong points:
- Aim of the study: In this study, the authors investigate the effect of geopolitical risk on the returns of the Information Technology, Communication Services, and Consumer Staples sector firms within the S&P 500 index.
- Research methodology and instruments: The authors use the event study methodology and perform more than 17,000 regressions to provide empirical evidence at sector-level that geopolitical risk leads to different responses across these 3 sectors.
- Results of the study: The authors’ findings are as follows: (a) the response of the Information Technology sector is negative for all event windows under study, except the one spanning 10 days prior to the geopolitical event and 10 days after; (b) the Communication Services sector has positive returns as a result of geopolitical events for all event windows, except the one from the geopolitical event date and 5 days after; and (c) the Consumer Staples sector shows a negative impact on geopolitical risk for all event windows except the one from the geopolitical event date and 5 days after, demonstrating a negative correlation to the Communication Services sector.
(B) Suggestions meant to improve your current manuscript:
Distinguished Authors I would kindly like to suggest inserting in your article a few ideas concerning the correlation between effects of the Covid-19 global crisis, sustainability, consumers’ choices and decisions, and the impact of geopolitical risk on the information technology, communication services, since these are key focuses these days. In this context, I had the chance to read a few interesting articles recently, among which I would like to mention: An Exploratory Study Based on a Questionnaire Concerning Green and Sustainable Finance, Corporate Social Responsibility, and Performance: Evidence from the Romanian Business Environment. J. Risk Financial Manag. 2019, 12, 162. DOI: 10.3390/jrfm12040162, link: https://www.mdpi.com/1911-8074/12/4/162.
Dear Authors, congratulations once again for your work and valuable insights reflected in the content of the manuscript, and I hope my comments will be of value to you!
Kind regards,
The Reviewer

Author Response
1.
Our study has discussed the impact of the global COVID-19 pandemic on economic performance, as seen in (Mazura et al. 2020; Bai et al., etc.). We’ve also added the reference (Popescu & Popescu 2019) you suggested for completeness in our paper.
2.
- We have reviewed our names.
- We have added new text in section 6.2 of our paper about future studies using distributed ledger technology, blockchains, and cryptocurrencies as follows: “Modern technologies such as machine learning analysis, distributed ledger technology, blockchains, and cryptocurrencies are currently being used to forecast the impact of certain risks on stock returns. Future research might include using these methods to investigate the impact of geopolitical risk on stock returns.”
- The practical implications of our research are already presented in section 6.1 of the paper as follows: “Our results also have some real-life and practical applications as follows: Companies within the Information Technology, Communication Services, and Consumer Staples sectors in the U.S. can intentionally study geopolitical events to determine how their stock prices could react whenever these uncertain events occur.
Furthermore, geopolitical risk is often a major cause of concern to company stakeholders who include policymakers, corporations, financial investors, politicians, etc. In particular, investors can decide to buy Information Technology stocks when a geopolitical event occurs due to the initial negative impact it has on the stock prices of the firms within this sector (since the Information Technology firms’ stocks will be trading at a discount compared to their future prices (for example 10 days after the event occurs)).”
Round 2
Reviewer 1 Report
Good job
Author Response
Thank you!
Reviewer 2 Report
The authors try to defence their viewpoint. However still there are two concenrs.
1. I do not agree that in the case when only 15-20% of abnormal returns are significant the reader must "see" it in the table which contains NA. The aware reader realizes that not all company shares react to each geopolitical event analyzed in the paper. It can be illustrated in a more summarized form instead of producing several pages of tables with NA, which are in fact boring. I strongly recommend to improve presentation of the results and to limit the number of pages. As for me in the tables only significant results should remain. The full structure of the results can be presented in some figures.
2. The information on robustness check is not convincing because it is desribed in 2-3 sentences only. There is no results comparison or differences between the approaches.
Author Response
Thank you for your comments! We have taken into accounts your comments for improving the paper.
Concerning comment 1 on redundant information stemming from a lot of N/A's, we have removed 3 tables from our text, since the other tables accurately represent the results.
Concerning comment 2:
1. It is well established that OLS estimation is advocated for event studies. Please see the articles we referred to for event studies.
2. You may agree that a robustness check is required in presence of heteroskedasticity. There was a typo in Table 1 which have been fixed. Instead of percent (%), it is now corrected by adding " multiplied by 100. Now, you can clearly see that the standard deviation between different windows does not vary much (less than 1) and kurtosis is very small. This is evidence for not having large extreme values. Please not also that due to separating windows, the effect of extreme value disappears. Hence, we do not see any reason for further robustness.